# SEARCH INSPIRED EXPLORATION FOR REINFORCEMENT LEARNING

## ABSTRACT

Exploration in environments with sparse rewards remains a fundamental challenge for reinforcement learning (RL). Existing approaches such as curriculum learning and Go-Explore often rely on hand-crafted heuristics, while curiosity-driven methods risk converging to suboptimal policies. We propose Search-Inspired Exploration in Reinforcement Learning (SIERL), a novel method that actively guides exploration by setting sub-goals based on the agent's learning progress. At the beginning of each episode, SIERL chooses a sub-goal from the *frontier* (the boundary of the agent's known state space) before the agent continues exploring toward the main task objective. The key contribution of our method is the sub-goal selection mechanism, which provides state-action pairs that are neither overly familiar nor completely novel. It assures that the frontier is expanded systematically and that the agent is capable of reaching any state within it. Inspired by search, sub-goals are prioritized from the frontier based on estimates of cost-to-come and cost-to-go, effectively steering exploration towards the most informative regions. In experiments on challenging sparse-reward environments, SIERL outperforms dominant baselines in both achieving the main task goal and generalizing to reach arbitrary states in the environment.

## 1 INTRODUCTION

Reinforcement learning (RL) holds the promise of enabling agents to master complex tasks by interacting with their environments. Yet applying RL in realistic domains remains challenging due to the combination of high-dimensional state–action spaces and sparse reward signals. In many environments, meaningful feedback is obtained only after completing long sequences of actions, making standard RL algorithms highly data-inefficient.

A central obstacle is the exploration–exploitation dilemma: agents must discover novel behaviors while simultaneously leveraging what they already know to make progress. Existing methods often overlook the problem of how an agent can *actively* direct its exploration to collect the most informative experiences (Amin et al.). Addressing this challenge is crucial for developing RL agents that learn more stably and scale to environments with delayed or infrequent rewards. We argue that progress requires shifting from agents that passively process environment feedback to those that deliberately seek out information in a principled way.

Several approaches have been proposed to address the challenge of exploration in sparse-reward reinforcement learning. Curriculum Learning (CL) introduces tasks of increasing difficulty to gradually shape agent behavior, but it relies on carefully hand-crafted difficulty metrics and is prone to negative transfer if the curriculum is poorly designed (Fang et al.; Liu et al.). Intrinsic motivation methods reward novelty or curiosity, encouraging the agent to seek unexplored regions of the state space. However, these methods are often a form of reward shaping, which can bias the learning process and lead to suboptimal policies. They are also susceptible to the "noisy-TV problem," where agents are distracted by stochastic but irrelevant features (Burda et al.; Ladosz et al.). Go-Explore (Ecoffet et al.) explicitly remembers and returns to promising states, but depends heavily on domain-specific heuristics and requires careful selection of interesting states.

Goal-Conditioned Reinforcement Learning (GCRL), particularly when combined with Hindsight Experience Replay (HER), offers another principled framework for overcoming these limitations by explicitly training agents to reach arbitrary states. We build on this paradigm to automatically

generate sub-goals that extend progressively farther from the start state. In doing so, our method effectively constructs a curriculum without requiring manually designed tasks or environments of varying difficulty.

In this work, we introduce *Search-Inspired Exploration* (SIERL), a novel approach that guides exploration by setting sub-goals informed by the agent's learning progress. Our main contributions are: 1. We propose a principled sub-goal selection mechanism that systematically expands exploration by defining a frontier of experience and prioritizing sub-goals using cost-to-come and cost-to-go estimates. 2. We design a novel *Hallway* environment that enables fine-grained control over exploration difficulty by varying the length of action sequences required to succeed. 3. We show that SIERL leads to more efficient exploration in discrete sparse-reward settings. 4. We present an empirical study that disentangles the contributions of individual components and identifies which mechanisms most effectively improve exploration for goal-conditioned agents. NEW

The remainder of this paper is organized as follows: section 2 reviews related literature; section 3 introduces the necessary preliminaries; section 4 details our algorithm; section 5 presents experimental results; and sections 6 and 7 discuss conclusions and future directions.

## 2 RELATED WORK

A wide range of exploration methods have been proposed for reinforcement learning (RL). These methods can be broadly categorized along several axes: whether they rely on extrinsic rewards or intrinsic exploration bonuses, employ memory or are memory-free, learn autonomously or from demonstrations, act randomly or deliberately (goal-based), or adopt an optimism-driven strategy (Amin et al.; Ladosz et al.).

**Novelty and optimism-based methods.** Novelty-bonus and optimism-based approaches encourage exploration by augmenting rewards with bonuses for visiting new or uncertain states. These methods are particularly useful in sparse-reward environments, where intrinsic signals provide more consistent feedback than delayed extrinsic rewards. In the bandit setting, the well-known Upper Confidence Bounds (UCB) algorithm balances exploration and exploitation by favoring actions with high value uncertainty (Auer et al.). In reinforcement learning, count-based techniques extend this principle by quantifying novelty through visitation counts over states or state–action pairs (**?**). Practical implementations rely on approximations such as hashing (Tang et al.), pseudo-counts (Ostrovski et al.), or elliptical episodic bonuses (Henaff et al.), all of which assign higher exploration bonuses to rarely visited regions of the state space. Pseudo-count methods in particular have demonstrated strong performance on hard-exploration benchmarks, notably achieving state-of-the-art results on Montezuma's Revenge (**?**). To avoid the limitations of explicit counting, Random Network Distillation (RND) (Burda et al.) introduces a scalable alternative: a predictor network is trained to match the outputs of a fixed, randomly initialized target network, and the prediction error serves as an intrinsic reward. Novel states typically yield higher prediction errors, thus guiding exploration toward regions where the agent's predictive model is least accurate. Broader novelty-driven methods extend beyond counts: optimistic initialization assumes unseen state–action pairs yield high returns, biasing agents toward exploration under the "Optimism in the Face of Uncertainty" principle (Treven et al.).

**Goal-based exploration methods.** Goal-based methods frame exploration as a deliberate process rather than relying on random or purely novelty-driven signals. By defining or generating explicit goals within the environment, these methods encourage the agent to learn policies that reach strategically important or unexplored states. This structured exploration typically involves three components: a mechanism for goal generation (e.g., sub-goals), a policy for goal discovery, and an overall strategy that coordinates exploration around these targets. Notable examples include Go-Explore (Ecoffet et al.), which achieves strong performance by explicitly remembering and returning to promising states before exploring further. Other approaches incorporate planning techniques, either within model-based RL frameworks (Hayamizu et al.) or by substituting policy search components with kinodynamic planners to better direct exploration (Hollenstein et al.). Planning Exploratory Goals (PEG) (Hu et al.) leverages learned world models to sample exploratory "goal commands" predicted to unlock novel states, from which the agent then explores.

**Frontier- and confidence-driven exploration.** Several recent methods refine goal-based exploration by explicitly reasoning about the frontier of reachable states or by incorporating measures of

confidence. Latent Exploration Along the Frontier (LEAF) (Bharadhwaj et al.) learns a dynamics-aware latent manifold of states, deterministically navigates to its frontier, and then stochastically explores beyond it to reach new goals. Temporal Distance-aware Representations (TLDR) (Bae et al.) exploit temporal distance as a proxy for exploration potential, selecting faraway goals to encourage coverage and training policies to minimize or maximize temporal distance as needed. In reset-free settings, Reset-Free RL with Intelligently Switching Controller (RISC) dynamically alternates between forward and backward exploration goals based on confidence in achieving them, effectively balancing task-oriented progress with revisiting initial states to diversify experience.

**Summary.** Together, these exploration strategies illustrate a steady evolution in RL: from simple count-based and novelty-driven approaches to increasingly structured methods that incorporate goal-setting, planning, and confidence-aware strategies. Novelty-based techniques provide intrinsic motivation to reduce uncertainty and expand coverage, but as a form of reward shaping, they can bias behavior and lead to suboptimal policies, in addition to being vulnerable to distractions such as the "noisy-TV problem." Goal-based approaches make exploration more intentional by defining explicit targets such as distant states, frontier boundaries, or strategically planned points. However, they often rely on brittle heuristics, handcrafted difficulty metrics, or domain-specific knowledge that limits generality. These drawbacks highlight an open gappx: how to design exploration methods that are both systematic and robust, capable of scaling beyond hand-tuned heuristics while ensuring that chosen exploratory targets remain novel but still reachable.

## 3 BACKGROUND AND PROBLEM SETUP

Exploration in reinforcement learning is especially challenging in environments with sparse rewards, where agents must solve long sequences of actions before receiving feedback. To formalize this setting, we focus on sequential decision-making problems with explicit goals, expressed through Goal Markov Decision Processes (GMDPs). This framework highlights the difficulty of discovering goals when reward signals are rare and emphasizes the role of the exploration–exploitation dilemma in guiding agent behavior.

### 3.1 THE HARD-EXPLORATION PROBLEM

Hard exploration problems are a direct consequence of sparse rewards, often exacerbated by large state and/or action spaces. When rewards are sparse, learning can be extremely slow because the agent wanders aimlessly for long periods without any signal to guide its behavior (Ladosz et al.). If the path to a reward is long and specific, random exploration strategies (like $\varepsilon$-greedy, with a small $\varepsilon$) are unlikely to find it in a reasonable amount of time. The agent might get stuck in local optima of familiar, non-rewarding behavior, or it might never encounter the critical states that lead to high rewards. Therefore, in these problems, we need more sophisticated exploration strategies that can intelligently seek out beneficial experiences for the agent to learn from.

### 3.2 GOAL MARKOV DECISION PROCESSES

We considered sequential decision-making problems that are formalized as Markov Decision Processes (MDPs). An MDP is defined by a tuple $\langle \mathcal{S}, \mathcal{A}, P_a, r, \gamma \rangle$, where $\mathcal{S}$ is a finite set of possible states, $\mathcal{A}$ is a finite set of actions available to the agent, $P_a\left(s' \mid s, a\right)$ is a function $\mathcal{S} \times \mathcal{A} \times \mathcal{S} \to [0, 1]$ that returns the transition probability defining the likelihood of transitioning to state $s'$ after taking action $a$ in state $s$; $r(s, a, s')$ is the reward function $\mathcal{S} \times \mathcal{A} \times \mathcal{S} \to [0, \infty)$ specifying the immediate reward received after a transition, and $\gamma \in [0, 1]$ is a discount factor that balances the importance of immediate versus future rewards. The primary objective of an agent in an MDP is to learn a policy $\pi : \mathcal{S} \to \mathcal{A}$ that maximizes the expected cumulative discounted reward ($\mathbb{E}_\pi[\sum_{t=0}^\infty \gamma^t r_{t+1}]$), often referred to as the value function.

A significant group of problems, particularly relevant in planning and many reinforcement learning applications, are goal-oriented tasks. Such tasks can be formalized using Goal-MDPs or *Shortest-path MDPs* (Bertsekas). In this formulation, there can be one or more designated goal states in the environment, and the agent's primary task at each point in time is to reach the current goal. $s_{\mathrm{G}} \subseteq \mathcal{S}$ is the set of all possible (absorbing) goal states. The reward structure in Goal MDPs is often adjusted to reflect this objective; a common setup involves a positive or zero reward upon reaching

---

**Algorithm 1:** SIERL Algorithm (abridged - details in Appendix A)

---

**Input:** agent, $s_\text{G}$
$frontier \leftarrow \{\}$                                          // Initialize frontier
**while** training **do**
    $s_\text{SG} = \texttt{get\_subgoal}(frontier)$                              // Get sub-goal
    **while** not timeout **do**
        $s' \leftarrow \text{Execute}(\pi(s, s_\text{G}))$
        $frontier.\text{update}(s, a)$                    // Insert (or not) in frontier
        **if** $\text{should\_switch}(s', s_\text{G})$ **then**
            $s_\text{SG} \leftarrow s_\text{G}$          // Sub-goal reached or early switching

---

a goal state and zero and small negative rewards (costs) for all other transitions. Thus, instead of the reward function $r(s, a, s')$, a cost function $c(a, s)$, a map $\mathcal{S} \times \mathcal{A} \rightarrow \mathbb{R}$, is used that specifies a cost for each action. Goal states $s_\text{G} \in \mathcal{S}$ can be absorbing, meaning $P_a(s_\text{G} \mid a, s_\text{G}) = 1$ for all $a \in \mathcal{A}$, and cost free, meaning $c(a, s_\text{G}) = 0$ for all $a \in \mathcal{A}$. This transforms the problem into one of finding an optimal path or policy for each goal, with the objective of achieving a desired terminal goal or condition. It has been shown that (partially observable) MDPs can be transformed into equivalent (partially observable) Goal MDPs (Bertsekas)

### 3.3 Exploration vs Exploitation dilemma

In Goal-MDPs with sparse rewards, the exploration–exploitation dilemma is particularly acute. Exploitation leverages past knowledge but offers little benefit early on, when goal rewards remain undiscovered. Exploration requires trying new actions and states without immediate payoff, often at high cost, but is essential for locating rare reward signals. The central challenge is to balance extensive exploration with eventual convergence on an optimal policy: without sufficient exploration, goals may never be found, but without exploitation, progress toward them cannot be consolidated.

## 4 Method

Our method, SIERL, introduces a principled way to perform deliberate exploration in reinforcement learning through goal-conditioned sub-goal setting. The key premise is that state–action regions become progressively less informative as they are explored more extensively: once the agent has learned accurate value estimates locally, further exploration in the same region yields diminishing returns. Instead, the agent should expand exploration toward novel but reachable states at the edge of its current knowledge, thereby extending the frontier of explored regions.

To achieve this, we employ a two-phase exploration process. In the first phase, the agent follows a goal-conditioned policy to reach selected frontier sub-goals, systematically expanding the boundary of explored states. In the second phase, the agent uses the experience gained in Phase 1 to explore efficiently toward the main task goal. This strategy combines systematic expansion with goal-directed exploration, ensuring both stable learning of an optimal policy for the task goal and improved generalization to alternative goals.

A pseudo-code description of SIERL is provided in algorithm 1, with full implementation details in Appendix A.

### 4.1 Two-Phase Exploration Strategy

Formally, we assume a goal-conditioned policy $\pi(a \mid s, g)$ that selects actions conditioned on the current state $s \in \mathcal{S}$ and a goal $g \in \mathcal{S}$. At the start of each episode, our method alternates between two phases: frontier-reaching exploration and main-goal exploration.

**Phase 1: Frontier Reaching and Expansion.** In the first phase, the agent is assigned a frontier sub-goal $s_\text{SG} \in \mathcal{F}$, where $\mathcal{F}$ denotes the frontier set extracted from the replay buffer $\mathcal{RB}$ (see subsection 4.2). The agent then executes the goal-conditioned policy $\pi(a \mid s, s_\text{SG})$ to deliberately reach $s_\text{SG}$. By incrementally selecting such frontier sub-goals, the agent systematically expands the

explored region of the state space in a curriculum-like fashion, while simultaneously improving its estimates of local dynamics and value functions.

**Phase 2: Main-Goal Exploration.** After reaching the frontier sub-goal $s_{\text{SG}}$, the agent transitions to the second phase and executes $\pi(a \mid s, s_{\text{G}})$, where $s_{\text{G}}$ denotes the main task goal. Starting exploration from $s_{\text{SG}}$ makes reaching $s_{\text{G}}$ more efficient, as the agent benefits from previously acquired experience near the boundary of known states.

**Phase Switching Strategy.** The transition between phases is governed by a hybrid deterministic–stochastic mechanism: 1. **Predefined horizons:** Each phase $i \in \{1, 2\}$ is assigned a maximum number of steps $H_i$, ensuring balanced allocation of exploration. 2. **Probabilistic early termination:** If during Phase 1 the agent encounters a novel state $s$ with a visitation count of $N_{\mathcal{RB}}(s) \leq N_{\text{thr}}$ ($N_{\text{thr}} = 1$ in our experiments), it may switch immediately to Phase 2 with probability $p_{\text{switch}} \in (0, 1)$, even if $H_1$ has not yet been exhausted.

### 4.2 Frontier Extraction

A critical aspect of our method is the identification of the frontier $\mathcal{F}$ from which the sub-goal is selected for the first phase. Those sub-goals are represented as state-action pairs $(s, a)$, instead of plain states. We initially filter the agent's past experiences from the replay buffer to select the best candidates. State-actions considered less novel or "very well known" are filtered out at this stage. In practice, we first rank the visited state-actions based on a familiarity score $F$ and exclude the *familiar* ones with a score above a threshold $F_\pi^{\text{thr}}$. The motivation is to maintain the focus of the exploration away from the increasingly more visited states, whose transitions will be occupying an increasingly larger part of the experience replay buffer. Formally, this filter can be expressed as:

$$\mathcal{F} = \{(s, a) \in \mathcal{RB} : F_\pi(s) < F_\pi^{\text{thr}}\} \tag{1}$$

The potential sub-goals are obtained from the same state-actions being inserted in the replay buffer $\mathcal{RB}$, which are filtered to maintain a continuously updated frontier, in the same manner an *Open list* and a *Closed list* is used in search. The frontier is populated with all state-actions that have been visited at least once and have a familiarity score below a threshold $F_\pi^{\text{thr}}$, as well as those actions on the newly states that have not yet been tried. More specifically, when a new state $s$ is visited for the first time, we insert all possible state-action pairs $(s, a_i)$ for all available actions $a_i \in \mathcal{A}$ into the frontier. For the edge-case when the frontier set obtained happens to be empty, we populate it with only the main goal, effectively turning that episode into a typical main-goal pursuit.   NEW

For each frontier state-action pair, the additional relevant information recorded is its visitation counts, $N(s, a)$, as well as its *familiarity score*, $F(s)$.

**Definition 1** (State Familiarity). *Let $\mathcal{RB}$ denote the replay buffer containing all past experiences of an agent, and let $N_{\mathcal{RB}}(s, a)$ be the visitation count of a state–action pair $(s, a) \in \mathcal{S} \times \mathcal{A}$ within $\mathcal{RB}$. The **familiarity** of $s$ with respect to $\mathcal{RB}$ is defined as*

$$F_{\mathcal{RB}}(s, a) = \frac{1}{1 + N_{\mathcal{RB}}(s, a)^{-1}}. \tag{2}$$

Such definition ensures that $F_{\mathcal{RB}}(s, a) \to 1$ as $(s, a)$ becomes frequent in $\mathcal{RB}$, and $F_{\mathcal{RB}}(s, a) \to 0$ when $(s, a)$ is rare. Besides state familiarity, we also define trajectory familiarity.

**Definition 2** (Trajectory Familiarity). *For a trajectory $\tau = \langle s_1, s_2, \ldots, s_k \rangle$ resulting from running goal-conditioned policy $\pi$ for a goal $s_k$, the familiarity of the terminal state $s_k$ is defined recursively as*

$$F_\pi(s_k) = \prod_{i=1}^{k} \frac{1}{1 + N_{\mathcal{RB}}(s_i)^{-1}}. \tag{3}$$

Assuming that we learn consistently and that policy $\pi$ conditioned on state $s_{k-1}$ results in trajectory $\langle s_1, s_2, \ldots, s_{k-1} \rangle$, we can calculate trajectory familiarity for state $s_k$ using the current state's visitation counts and trajectory novelty the previous one: $F_\pi(s_k) = \frac{1}{1+N(s_k)^{-1}} F_\pi(s_{k-1})$.

Motivation for such a definition is that when reaching a sub-goal, if the current policy succeeds in reaching it through familiar states, that should indicate that the agent has mastered reaching

that state and can focus on further states. Using products in the calculation ensures the balance of the influence of trajectory length and the effect of familiarity of individual states. This strategy, particularly when combined with the probabilistic early switching mechanism, ensures that while the frontier is gradually populated with states near the expanding boundary of the familiar region, the agent concurrently gains experiences in states that are adjacent and relevant to each chosen sub-goal. This promotes a more consistent and thorough exploration.

### 4.3 Sub-goal Selection

The remaining state-action pairs that form the frontier $\mathcal{F}$ after the filtering are then ranked and prioritized. This prioritization is determined by minimizing a combination of the following cost factors:

**Novelty Cost** $c_{\mathrm{n}}$ This cost penalizes more novel states, thereby favoring those familiar states that are more visited while still not overly familiar (since they have passed the initial filtering stage). This is based on the idea that the agent should first focus on mastering sub-goals it already practices before continuing further. Additionally, states visited extremely infrequently might be outliers or part of highly stochastic regions not yet suitable for directed exploration.

**Cost-to-Come** $c_{\mathrm{c}}$ (from the initial state to the sub-goal): This is estimated directly using the learned Q-values, representing the expected cumulative reward (or cost, in our negative reward setting) to reach the potential sub-goal from the episode's starting state, calculated as $\max_{a \in \mathcal{A}} Q(s_{\mathrm{I}}, s_{\mathrm{SG}})$.

**Cost-to-Go** $c_{\mathrm{g}}$ (from sub-goal to main goal): This is the estimated cost from the potential sub-goal to the ultimate task goal, again derived from the learned Q-values as $\max_{a \in \mathcal{A}} Q(s_{\mathrm{SG}}, s_{\mathrm{G}})$.

Thus, the score used for prioritizing the filtered goals can be formulated as the sum of each one's cost-to-come $c_{\mathrm{c}}$ and cost-to-go $c_{\mathrm{g}}$, weighted by $\mathbf{w}$, multiplied by the novelty cost, which is also weighted with a weight-exponent $w_{\mathrm{n}}$. The sub-goal is sampled with probability assigned by applying a softmin to this set of scores for the frontier state-actions:                                                    NEW

$$P\big((s,a) = (s_{\mathrm{SG}}, a_{\mathrm{SG}})\big) = \operatorname*{softmin}_{(s,a) \in \mathcal{F}} \big(c_{\mathrm{n}}(s,a)^{w_{\mathrm{n}}} \mathbf{w}^{\mathsf{T}} \mathbf{c}(s)\big). \tag{4}$$

Where:

$$c_{\mathrm{n}}(s,a) = \sigma(z(-N(s,a))),$$
$$\mathbf{w}^{\mathsf{T}} = \begin{bmatrix} w_{\mathrm{c}} & w_{\mathrm{g}} \end{bmatrix},$$
$$\mathbf{c}(s) = \begin{bmatrix} c_{\mathrm{c}}(s) & c_{\mathrm{g}}(s) \end{bmatrix}^{\mathsf{T}},$$
$$z(x) = \frac{x - \mathbb{E}[X]}{\mathrm{Var}(X)}.$$

Thus, the state with the optimal combined score is selected as the next sub-goal for the agent in Phase 1.

## 5 Experiments

For our experiments we aimed to set up situations which require deliberate exploration and a more thorough coverage of the state space to be solved. We strove to answer the following: (a) Does SIERL enable consistently succeeding in environments where goal discovery is non-trivial? (b) In which cases and in which aspects SIERL is more promising than its competitors? (c) Which components enable SIERL to perform well?

### 5.1 Setup

Our experiments are designed to evaluate the performance of SIERL in scenarios that demand deliberate exploration and a comprehensive understanding of the state space. We use discrete state and action environments, where goal discovery is non-trivial due to sparse rewards and deceptive rewards from "trap" obstacles. We adjusted the rewards such that a signal of -1 is given for each step, and a reward of 0 upon reaching the goal, effectively turning the task into a shortest path

problem. During evaluation, we run 10 main-goal reaching episodes as well as 10 random-goal reaching episodes for each method, reporting the mean success rate and standard error, in order to capture the methods' capacity to generalize while learning with a specific goal. All methods are run with 10 seeds to account for variance. The environments used are a subset of the MiniGrid framework (Chevalier-Boisvert et al.) and the experiments were set up using RLHive (Patil et al.).

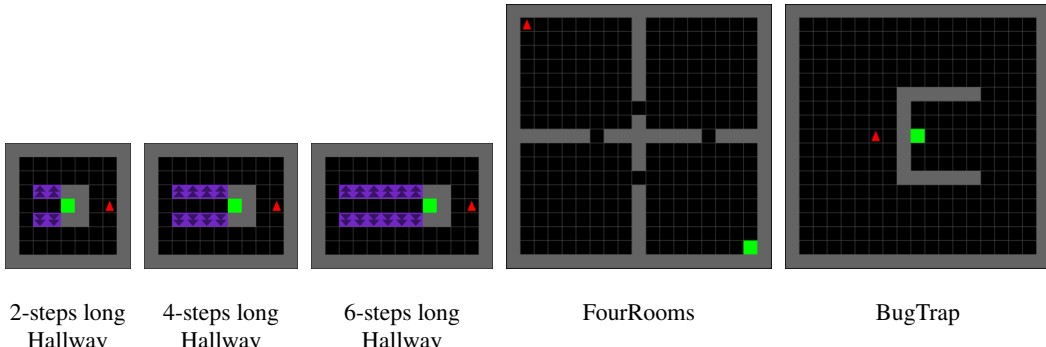

| 2-steps long Hallway | 4-steps long Hallway | 6-steps long Hallway | FourRooms | BugTrap |

Figure 1: The MiniGrid room variants.

These MiniGrid environments are minimalistic 2D grid worlds set up with a discrete action space representing moving left, right, up, or down. The state space is fully observable, with the agent's goal being to reach a specific static goal state. The agent's observation is a grid containing information about its location, the walls/obstacles, and the goal location. We specifically used several custom-made variants of a *Hallway* room, the *FourRooms* room, and a typical *BugTrap* room.

**Hallway variants**: These are challenging environments, containing a hallway flanked with "slippery" unidirectional tiles along the sides, as shown in Figure 1. The goal lies at the end of each corridor and the agent is required to perform a precise (albeit repetitive) sequence of actions to reach its end.

**BugTrap**: In this room, the agent has to navigate around a concave enclosure to reach the goal on the other side. Being more open requires the agent to progressively explore a larger region of state-action space until reaching the goal.

**FourRooms**: The agent is required to navigate from one corner of a square space comprised of 4 rooms to the diagonally opposite corner, through doors between the rooms. Reaching arbitrary locations in this more segregated space is a harder task than in the other cases.

## 5.2 RESULTS

**Main goal success rates:** The evaluation performance in the success rate for reaching the main goal is shown in the upper part of Figure 2. In all three Hallway variants SIERL performed on par with the most competitive baselines, such as Novelty bonuses, while outperforming HER, and Q-Learning. More details about the baselines are presented in Appendix C. Specifically, on the small enough 2- and 4-step long Hallways, Random-goals Q-learning performs similarly as well; however, its performance is hampered on the larger 6-step long variant, following closely behind that of Novelty bonuses, whose performance is also impacted, albeit to a smaller degree. Nonetheless, SIERL is always able to discover and learn the main goal for all seeds.

In FourRooms, SIERL performs comparably well to HER but less so compared to Novelty bonuses. It is notable that succeeding in such an environment requires systematic coverage of the state-action space, which is accomplished via intrinsic rewards but not by relying solely on random exploration. This indicates SIERL is able to learn on the less accessible parts of the state space and, contrary to Novelty bonuses, it accomplishes that without tampering with the reward signal, but rather by guiding the agent's exploration and thus adjusting the experience distribution to improve learning.

**Random goal success rates:** The success rate during evaluation for reaching uniformly sampled random goals is shown in the lower part of Figure 2. In all cases, the only methods capable of solving for arbitrarily set goals are SIERL and Random-goals Q-learning. In the smallest 2-step Hallway

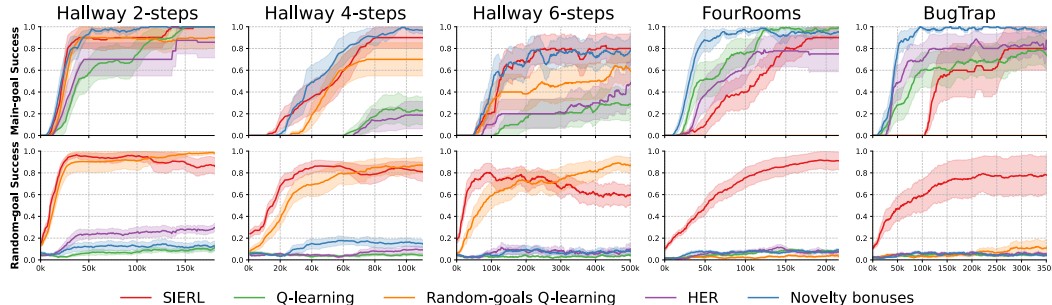

Figure 2: Main-goal (top row) and random-goal (bottom row) performance for the Hallway variant in columns. SIERL achieves a remarkable performance for both criteria at the same time, matched by no other method.

variant SIERL outperforms Random-goals, while this is less pronounced in the harder variants, where the latter continues improving at a slower pace. Notably, on the two larger environments of BugTrap and FourRooms, SIERL clearly outperforms every baseline. Arguably, the capacity of the Q-value network's architecture imposes a limit on the simultaneous learning of both a wide range of goals and a specific main goal. Both SIERL, and a random-goal focused method leverage this capacity better than the other methods, while each trading off main-goal and random-goal focus in different degrees.

This behavior can possibly be attributed to having a more diverse goal distribution in the experiences' transitions. It is also notable that SIERL is able to reach, learn on, and set sub-goals from a larger portion of the state space than HER, without relying on augmenting its experiences. The more systematic training with goals in a gradually expanding subset of the state space might prove beneficial to such generalization, provided those sub-goals are feasible, while at the same time managing to consistently learn to reach a main goal.

## 5.3 ABLATION STUDIES

To identify the crucial components enabling SIERL's success, we performed a series of ablation studies on the most challenging environment variant, to observe performance differences in several aspects. The main-goal, random-goal, and sub-goal (during training) performance of all variants was examined. The core ideas of SIERL are: guiding the exploration by gradually expanding the state space's well-learned region, while pursuing sub-goals towards the most promising direction of expansion of the region's frontier.

The first aspect we ablated was the early switching mechanism of the first phase of exploration. This way, the agent's experience gathering when pursuing sub-goals will extend without constraints further past the frontier of the familiar region, which contains the prospective sub-goals. Subsequently, focused on the contribution of the frontier extraction from experience filtering using the *familiarity* measure. By removing the extraction, the state-actions of which will be prioritized (the frontier) consist now of the complete set of experiences the agent has gathered, including all frequently tried state-actions. Lastly, we ablated the prioritization strategy of SIERL. In this case, the filtered states are not subjected to any scoring, and the sub-goal is picked at random with uniform probability. The aim is to evaluate the effectiveness of the prioritization strategy.

The ablation experiments' results are shown in Figure 3. While random-goal performance seems unaffected for all variants, barring one, all of them exhibit a negative impact on either the ramp-up time or stability in reaching the main goal. Specifically, removing frontier prioritization for selecting sub-goals results in notably worse performance on learning for the main goal in FourRooms. Likewise, ablating early-switching slightly worsens main-goal performance, although random-goal performance appears more stable on 6-steps Hallway. The seemingly better case of ablating the frontier filtering shows better random-goal performance, which is expected as the agent is consistently provided with a wider range of goal-conditioned experiences; however, it struggles to consistently learn on the main-goal. These observations further reinforce our understanding that SIERL demon-

strates the capacity to stably balance learning on both types of goals with the same sample-efficiency as other competitive single-goal focused methods. NEW

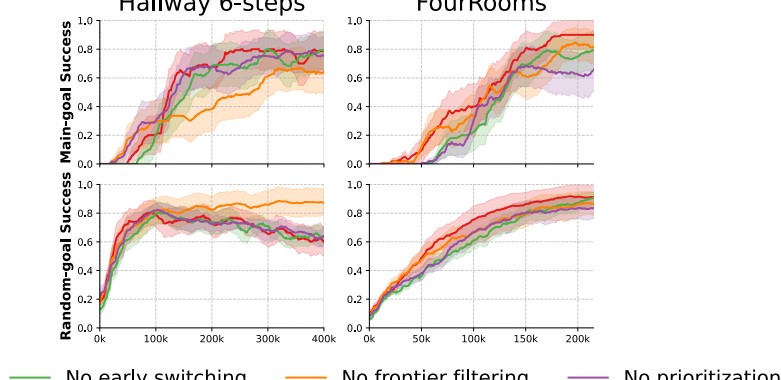

Figure 3: Success rate for reaching: the main goal (top), and random goals (bottom) for ablated variants, in 6-steps Hallway (left) and FourRooms (right). Most notably, removing frontier filtering or prioritization worsens SIERL's main-goal success, while removing early switching shows a smaller negative influence.

## 6 CONCLUSION

In this work, we presented SIERL, a method for Search Inspired Exploration in RL. This method is based on the principle that by gradually expanding the frontier of the explored region of the state space using sub-goal setting, the agent is able to efficiently cover the state space while learning a robust goal-conditioned behavior. In discrete settings, SIERL exhibits competitive performance in reaching the main goal, while simultaneously learns to reach any other state within range of its familiar region; a property all of the baselines lack. NEW

We demonstrated that this method is particularly suitable for hard exploration environments where getting from start to goal requires strictly executing a sequence of actions. Through ablation studies, we have shown that keeping the exploration's first phase within the familiar region (with early switching) and by selecting its sub-goals by prioritizing states in a frontier, which is extracted by filtering the agent's experiences, are all crucial components for SIERL's success.

## 7 LIMITATIONS

In its present implementation SIERL is limited to discrete-state action spaces as it relies on visitation counts to define the notions of novelty. A measure that can provide a more generic notion of novelty on any location in the state-action space, usable in continuous state-action spaces as well, can enable SIERL to be used on a wider range of problems. This could be done by adopting one of the approximate methods for pseudo-counts. We believe that regardless of the way in which the visitation counting and novelty is replaced, the *familiarity* notion is preserved. The current implementation is also limited by the capacity of the replay buffer, depending on the state-action space size, dimensionality, and discretization scheme.

Although they are intuitive, several hyper-parameters are pre-determined and environment-dependent, providing opportunities for exploring more environment-agnostic definitions and adaptations. Determining the familiarity threshold is dependent on the size of the state-action space and the distance between start and goal. A broader concept of familiarity would be linked more directly to the degree the agent has learned about parts of the state-action space, rather than assuming this to be so based on experience counting.

Similarly, there is room for improving the phase lengths and phase-switch timing. While also presently environment-dependent and fixed, these parameters can benefit from an implementation

more reliant on the agent's learning at each point during training. Ideally, selecting a new goal and determining the right time to do so should be done, aiming to balance pursuing novelty and providing "practicing" for a goal-conditioned agent.

## LLM USAGE STATEMENT

During the preparation of this document, an LLM was used for grammar, punctuation, and wording improvements. The core ideas, research, and conclusions are the authors' own.

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

## A   SIERL IMPLEMENTATION

SIERL, presented in algorithm 2, implements the two-phase exploration strategy detailed previously. The core of the algorithm operates in a continuous loop. Each iteration begins with the goal-setting and switching logic, including initialization of the environment, yielding an initial state $s_\mathrm{I}$ and the main task goal $s_\mathrm{G}$. After a reset, the agent's state is set to $s_\mathrm{I}$ and its current sub-goal, agent.*goal*, is set to $s_\mathrm{G}$, while the trajectory step counter $t$ is initialized to zero at the start of each phase.

At the start of each iteration is the decision to switch sub-goals, determined based on the following conditions: First, if the agent's current state $s'$ matches the main goal $s_\mathrm{G}$, then a reset is performed and a sub-goal is generated, thus starting Phase 1. The sub-goal is selected. Otherwise, if the current state $s'$ matches the state component of the current agent.*goal*, then the Phase 1 sub-goal was just reached and it is time to move to Phase 2 by directing the exploration towards $s_\mathrm{G}$. If that is not the

case either, then if it is time for an early switch from Phase 1, or if the current trajectory length $t$ has reached a predefined maximum $M$, then the transition from Phase 1 to 2 takes place likewise.

Early switching from Phase 1 is performed by calling early_switch() (algorithm 4) which samples a random variable to determine whether to switch based on a predefined switching probability as follows: if the agent encounters a state $s'$ it has never visited before (i.e., agent.$visitations[s'] ==$ 0), a switch occurs with a probability $P_{switch}(state\_is\_novel = \texttt{true})$. Otherwise, if the agent's state is not novel, the probability to switch to the next phase is $P_{switch}(state\_is\_novel = \texttt{false})$ (typically lower or zero).

The sub-goal for Phase 1 is obtained by calling get_subgoal() (algorithm 3), which filters and prioritizes states from the replay buffer. This is achieved by first calling the get_frontier() method (algorithm 5) to obtain a list of candidate frontier state-action pairs. By iterating through the agent's replay buffer, the pairs $(s, a)$ whose familiarity agent.$familiarity[s, a]$ is higher than the maximum threshold $F_\pi^{thr}$, and those whose counts are above the minimum allowed percentile threshold $P_{10}(N)$. Subsequently, for each candidate frontier $(s_f, a_f)$ pair from this list, a cost is calculated as was described in subsection 4.2. The new sub-goal is then sampled, biased towards the state-action pair with the minimum calculated cost (e.g., by using a softmin distribution over the costs).

Subsequently, the agent selects an action $a$ based on its current state $s$ and the active agent.$goal$ using its goal-conditioned policy (e.g., $\varepsilon$-greedy). Upon executing the action, the environment transitions to a new state $s'$ and provides a reward $r$. The agent then updates its internal model, its Q-values or policy, using the experience $(s, a, r, s', \texttt{agent}.goal)$, as well as its familiarity for the last state-action with update_familiarity(), and the step counter $t$ for the current phase is incremented. In this update step the batch of randomly sampled experiences can contain transitions with either the main goal or any other previous frontier sub-goal.

Finally, the agent's current state $s$ is replaced by $s'$, and the loop continues. This interplay between pursuing generated sub-goals (Phase 1) and the main task goal (Phase 2), guided by the frontier extraction and prioritization logic, allows SIERL to systematically expand the familiar region, while moving towards the main goal.

---

**Algorithm 2:** SIERL Algorithm

**Input:** Agent agent; Environment env

**while** true **do**
    **if** $s_G = \texttt{null} \vee s = s_G$ **then**
        $s, s_G \leftarrow \texttt{env.reset}()$; $t \leftarrow 0$; $f \leftarrow 0$                 // episode reset
        agent.$goal$ = agent.get_subgoal($s$)
    **else if** $s = \texttt{agent}.goal.s$ **then**
        agent.$goal \leftarrow s_G$; $t \leftarrow 0$            // sub-goal reached, switch
    **else if** $(\texttt{agent.early\_switch}(s) \wedge \texttt{agent}.goal \neq s_G) \vee (t \geq M)$ **then**
        agent.$goal \leftarrow s_G$; $t \leftarrow 0$       // unwanted exploration or timeout
    **else**
        $a \leftarrow \texttt{agent}.\pi(s, \texttt{agent}.goal)$
        $s', r \leftarrow \texttt{env.step}(s, a)$
        $\texttt{agent}.\pi.\texttt{update}(s, a, r, s', \texttt{agent}.goal)$
        agent.$familiarity[s, a] \leftarrow$ update_familiarity($s, a, f$)
        $f \leftarrow \texttt{agent}.familiarity[s, a]$
        **foreach** $a' \in \mathcal{A}$ **do**                // add all possible $(s', a)$
            agent.$frontier$.insert($s', a'$)
        **end**
        **if** $f < F_\pi^{thr}$ **then**             // exclude too familiar $(s, a)$
            agent.$frontier \leftarrow$ agent.$frontier \cup \{(s, a)\}$
        **end**
        $t \leftarrow t + 1$
        $s \leftarrow s'$
    **end**
**end**

---

**Algorithm 3:** `get_subgoal()` method

---

**Input:** Current state: $s$

$costs \leftarrow \{\}$
**foreach** $(s_{\mathrm{f}}, a_{\mathrm{f}}) \in \mathtt{agent.get\_frontier()}$ **do**
$\quad c \leftarrow \sigma(-z(\mathtt{agent}.visitations[s_{\mathrm{f}}, a_{\mathrm{f}}]))^{w_{\mathrm{n}}}$         `// novelty cost`
$\quad\quad\quad \cdot\sigma(z(w_{\mathrm{r}}\mathtt{agent}.Q(s, s_{\mathrm{f}})+$         `// cost-to-reach`
$\quad\quad\quad\quad\quad w_{\mathrm{c}}\mathtt{agent}.Q(s_{\mathrm{I}}, s_{\mathrm{f}})+$       `// cost-to-come`
$\quad\quad\quad\quad\quad w_{\mathrm{g}}\mathtt{agent}.Q(s_{\mathrm{f}}, s_{\mathrm{G}})))$        `// cost-to-go`
$\quad costs \leftarrow costs \cup \{c\}$
**end**
$subgoal \leftarrow \mathrm{sample}(\mathrm{softmin}(costs))$     `// sample based on minimum cost`
**return** $subgoal$

---

**Algorithm 4:** `early_switch()` method

---

**Input:** Current state: $s$
        Switching probabilities: $P_{\mathrm{switch}}$

$state\_is\_novel \leftarrow \mathtt{agent}.visitations[s] == 0$
$early\_switch \leftarrow \mathrm{random}() < P_{\mathrm{switch}}(state\_is\_novel)$
**return** $early\_switch$

---

**Algorithm 5:** `get_frontier()` method

---

**Input:** Familiarity threshold: $F_{\pi}^{\mathrm{thr}}$

$frontier \leftarrow \mathtt{agent}.open\_list$
**foreach** $(s, a) \in frontier$ **do**
$\quad$ **if** $\mathtt{agent}.familiarity[s, a] > F_{\pi}^{\mathrm{thr}}$ **then**
$\quad\quad frontier \leftarrow frontier \setminus \{(s, a)\}$    `// exclude too familiar (s,a) pairs`
$\quad$ **end**
**end**
**foreach** $(s, a) \in frontier$ **do**
$\quad$ **if** $\mathtt{agent}.visitations[s, a] < P_{10}(N)$ **then**
$\quad\quad frontier \leftarrow frontier \setminus \{(s, a)\}$      `// exclude too novel (s,a) pairs`
$\quad$ **end**
**end**
**return** $frontier$

---

## B  AGENT DETAILS

NEW

All agents for our experiments use DQN (Mnih et al.) as the base agent. The corresponding hyperparameters commonly used in all experiments are shown in Table 1. The additional hyperparameters for SIERL are shown in Table 2.

| Hyperparameter | Value |
|---|---|
| $c_\mathrm{n}, c_\mathrm{c}, c_\mathrm{g}$ | $[1.5, 1.0, 0.5]$ |
| $P_\mathrm{switch}$ | 100% |
| $H_1, H_2$ | [episode length $- 1, 1$] |
| softmin temperature | 0.5 |
| Q-Network LR | $3 \times 10^{-4}$ |
| Optimizer | Adam |
| Target hard update frequency | episode length |
| Initial Collect Steps | 128 |
| Batch Size | 128 |
| Discount Factor | 0.95 |
| $\epsilon$-greedy init value | 1.0 |
| $\epsilon$-greedy end value | 0.1 |
| $\epsilon$-greedy decay steps | 20,000 |
| Seeds | 18995728, 64493317, 49789456, 22114861, 50259734, 99918123, 71729146, 10365956, 83575762, 35232230 |

Table 1: Common hyperparameters for all environments.

| | Hyperparameter | Value |
|---|---|---|
| Hallway 2-steps | $F_\pi^\mathrm{thr}$ | 0.9 |
| | Q-Network | Conv($3 \times [16]$) FC($[16]$) |
| | Replay Buffer | 100,000 |
| | episode length | 150 |
| Hallway 4-steps | $F_\pi^\mathrm{thr}$ | 0.9 |
| | Q-Network | Conv($3 \times [16]$) FC($[16]$) |
| | Replay Buffer | 100,000 |
| | episode length | 300 |
| Hallway 6-steps | $F_\pi^\mathrm{thr}$ | 0.95 |
| | Q-Network | Conv($3 \times [16]$) FC($[16]$) |
| | Replay Buffer | 100,000 |
| | episode length | 400 |
| FourRooms | $F_\pi^\mathrm{thr}$ | 0.8 |
| | Q-Network | Conv($7 \times [16]$) FC($[16]$) |
| | Replay Buffer | 300,000 |
| | episode length | 500 |
| BugTrap | $F_\pi^\mathrm{thr}$ | 0.7 |
| | Q-Network | Conv($7 \times [16]$) FC($[16]$) |
| | Replay Buffer | 300,000 |
| | episode length | 500 |

Table 2: Environment-specific hyperparameters.

### B.1  EXPERIENCE AND FRONTIER MANAGEMENT

SIERL's frontier is used in the same way as the Open list is used in search algorithms. However, although the Open list is often formulated as being updated on the fly, with entries being added and removed during every step, SIERL's frontier population is optimized to minimize excessive list manipulation actions, and redundant data storage to optimize memory use. In practice, the frontier is obtained from the Replay Buffer in a "lazy" manner whenever it is required in order to obtain a sub-goal.

To that end, a separate list is maintained containing a single entry for each unique state in the Replay Buffer, along with additional metadata, and it is the only SIERL's data structure that is being updated after each agent's step. This is a dictionary that contains the aforementioned unique state-action arrays (converted to hash-able data types) as its entries' indices or keys, and metadata such as visitation counts and familiarity values for its entries' values. After each step, the newly inserted transition is used to update this list. This entails updating the transitions' involved states' metadata, inserting new entries for newly visited states if needed, as well as removing entries that were last pushed out of the Replay Buffer due to new transition insertions.

Therefore, using this dictionary that contains all relevant metadata about the agent's experiences in an easily traversable format, the frontier is generated on the fly. By iterating once over it, filtering out all entries with a familiarity above the specified threshold, and inserting copies of the remaining to a new object. Thus, owing to it's "lazy" evaluation (applying filters while traversing long

pre-populated lists) SIERL's frontier is obtained with as little computation and memory usage as possible. The filtering algorithm is shown in pseudo-code in algorithm 5.

## C  BASELINES

To evaluate SIERL's performance, we compared it's performance against four baseline methods with different exploration strategies. All baselines, like SIERL, are built upon a goal-conditioned framework, meaning the agent's policy takes both the current state and a desired goal as input. This shared structure allows us to isolate the impact of different exploration and goal-selection strategies.

The first baseline is a Q-learning agent with $\varepsilon$-Greedy exploration, which is a well-established method for discrete state-action spaces. The second baseline is Q-learning augmented HER. HER addresses the sparse reward problem in goal-conditioned RL by re-purposing failed episodes by "re-labeling" their transitions. When an agent fails to reach its intended goal, HER modifies a portion of the transitions from that episode, but with their goal set to the state that the agent actually reached. This turns both phases in a failed episode into a successful trajectory from a different perspective, providing a less sparse reward signal and making learning more efficient.

The third baseline extends the Q-learning agent by adding a *novelty-based exploration bonus*. $\varepsilon$-Greedy exploration is still used, but its goal-sampling strategy is guided by *visitation counts*. During training, the agent keeps track of how many times it has visited each state. It then augments the reward signal by adding a bonus inversely proportional to the visitations for the visited state. This intrinsic motivation encourages the agent to explore new and under-explored regions of the environment, a strategy present in many state-of-the-art exploration methods. It is expected that after a long enough exploration, the bonuses diminish and the value function converges to the true values (Tang et al.). This method was selected as a representative of the intrinsic reward family of methods. All such methods (pseudo-counts, intrinsic curiosity modules, random network distillation error) define a reward bonus that is high if the current state is different from the previous states visited by the agent, and low if it is similar (Henaff et al.). These methods approximate the novelty each to a different degree. Using counts (when possible) provides the most precise way to quantify and reward novelty, compared to e.g. approximating surprise with RND. For this reason, and given that count-based novelty bonuses has shown good performance in discrete state-action settings, we used this method as an upper bound for all of them.

Our fourth baseline is another variation of Q-learning that uses the same setup but is trained exclusively on *random goals*. Unlike our other baselines and SIERL, this agent does not have a fixed "main goal" throughout the training. It samples all training goals uniformly at random from the state space. This serves as a critical benchmarking study, as it helps us understand the importance of goal-centric exploration. By comparing SIERL to this baseline, we can quantify the benefit of a method that deliberately focuses on gradually discovering and achieving specific, potentially difficult, goals as opposed to just exploring the entire state space uniformly at random.

## D  ADDITIONAL RESULTS

In addition to the main experiments presented in the main body of this work, we have conducted additional experiments such as sensitivity studies on SIERL's hyperparameters, running with probabilistic transitions, and additional harder environments with a maze-like 9 room arrangement. Our findings are presented in this appendix.

### D.1  SENSITIVITY STUDIES

As can be seen from the results in figures 4, 5, 6, and 7. SIERL demonstrated robust performance. Our results indicate that adjusting the weights, $w_{\mathrm{n}}, w_{\mathrm{c}}, w_{\mathrm{g}}$ , has a negligible impact on outcomes. However, the algorithm is moderately sensitive to the softmin temperature (specifically at larger values) and more sensitive to the familiarity threshold.

As expected, performance diminishes with a higher softmin temperature; this results in a "softer" distribution and near-uniform random prioritization of sub-goals, which nullifies the benefits of this method. Regarding the familiarity threshold: an excessively high value slows frontier expansion and

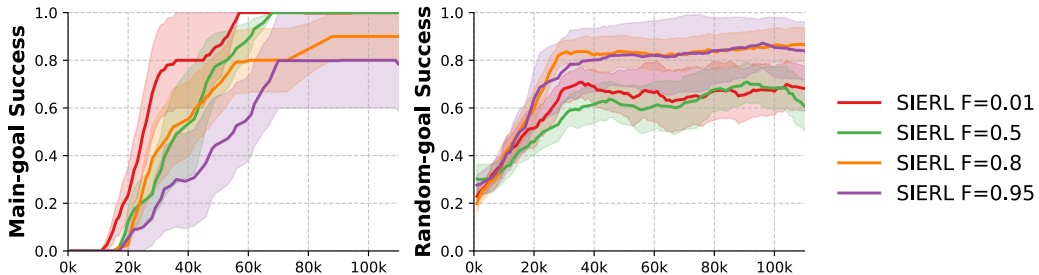

Figure 4: Success rate for the main-goal and random-goals in 4-step long Hallway for SIERL with varying familiarity thresholds, $F_\pi^{\mathrm{thr}}$.

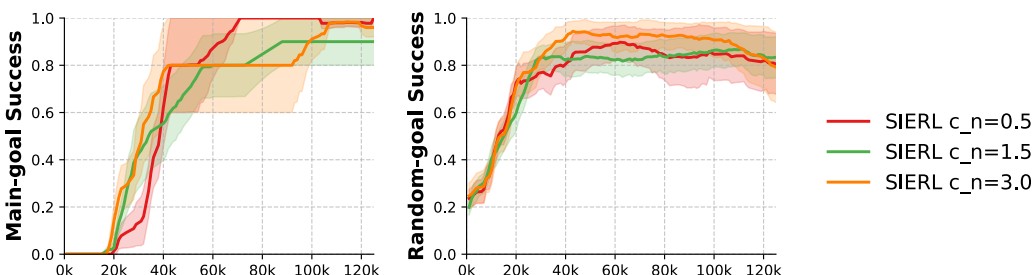

Figure 5: Success rate for the main-goal and random-goals in 4-step long Hallway for SIERL with varying novelty weights, $w_{\mathrm{n}}$.

solution-trajectory discovery, though it maintains a stable random-goal success rate. Conversely, a lower threshold promotes aggressive novelty-seeking at the expense of "practicing" in familiar regions. This significantly reduces the random-goal success rate—while leaving main-goal success largely unaffected—unless the environment is sufficiently difficult that such aggressive expansion disrupts the learning process entirely."

## D.2 PROBABILISTIC TRANSITIONS

In order to evaluate SIERL's performance outside of deterministic settings, we conducted an additional trial with probabilistic transitions. In particular, we used "slippery actions" such as those used in other discrete state-action settings in the literature. In this case, when the agent chooses an action, there is a small probability of executing another adjacent one. E.g., choosing to take the action "up" will result in the action taken being up with 80% chance, while there is a 10% chance of it being "right", and 10%chance of it being "left". The resulting performance of SIERL and all original baselines can be seen in Figure 8 It can be observed that SIERL is outperforming all baselines here as well. When compared to Figure 2, all baselines' performance is observed to be more notably impacted by the noisy actions than SIERL's. In main-goal performance SIERL is now performing better that novelty bonuses, the leading baseline, while still achieving random-goal success-rate above that of Random-goals Q-learning. Other than a slight decrease in main-goal success rate, no other significant impact is noticeable.

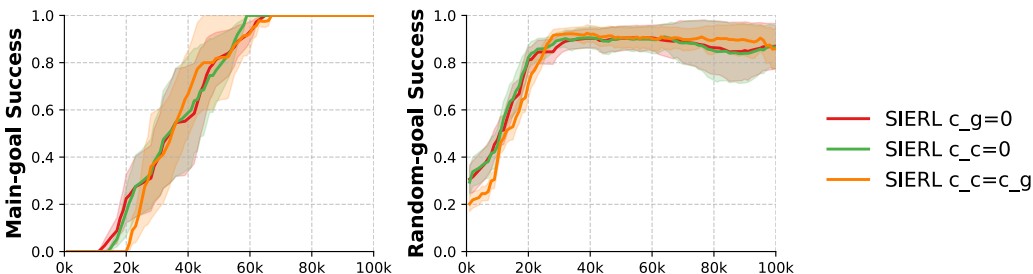

Figure 6: Success rate for the main-goal and random-goals in 4-step long Hallway for SIERL with varying path costs' weights, $w_c$ and $w_g$.

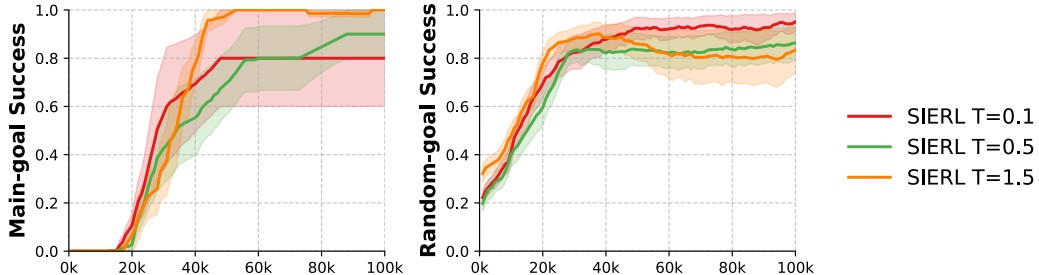

Figure 7: Success rate for the main-goal and random-goals in 4-step long Hallway for SIERL with varying softmin temperatures.

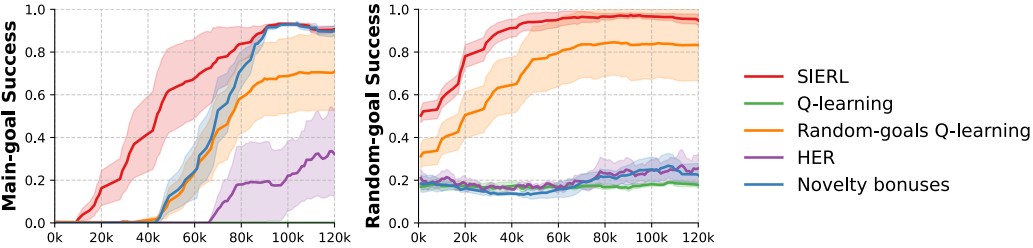

Figure 8: Success rate for the main-goal and random-goals in 4-step long Hallway with probabilistic transitions based on "slippery actions".

### D.3 THE MULTIROOM AND DOORKEY ENVIRONMENTS

We conducted additional experiments on more challenging environments, including the nine-room variants of MultiRoom and DoorKey, as shown in Figure 9. These domains feature larger spatial structure, longer horizons, and more complex navigation dependencies, providing a stronger test of scalable exploration and generalization.

The results in Figure 10 show that, in terms of main-goal performance, SIERL matches the strongest novelty-bonus baselines, which explore aggressively in these settings. Importantly, SIERL achieves substantially better random-goal generalization, demonstrating that the frontier-based curriculum supports broader mastery of the environment rather than focusing solely on the shortest path to the main goal.

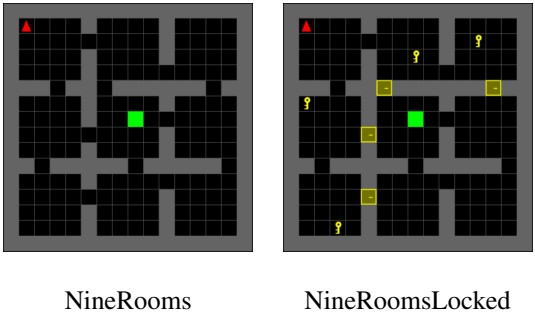

NineRooms          NineRoomsLocked

Figure 9: The nine rooms variants of MultiRoom and DoorKey.

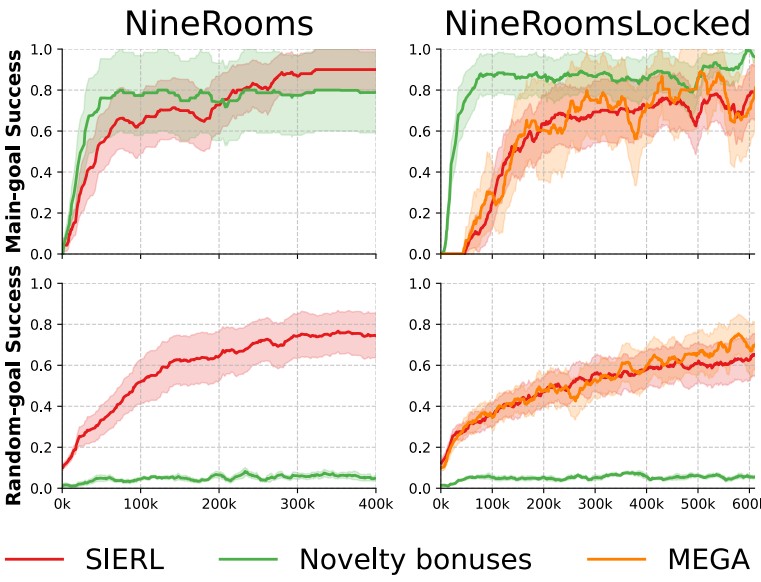

Figure 10: Success rate for the main-goal and random-goals in nine rooms environment from MultiRoom and DoorKey for SIERL and Novelty Bonuses.

### D.4 COVERAGE

We additionally provide a coverage analysis to visualize how exploration progresses over time. Figure 11 shows state visitation counts in NineRoomsLocked for the novelty-bonus baseline (top) and for SIERL (bottom) at 2k, 100k, 200k, 300k, 400k, 500k, and 600k environment steps. Visitation frequencies are shown on a logarithmic scale. The agent starts in the top-left room, and the main goal is located in the center room.

Under SIERL, coverage expands outward in a structured manner. After the agent first reaches the main goal (around 100k steps), the frontier continues to grow, and regions farther from both the initial position and the goal receive progressively fewer visits. This reflects the incremental broadening of mastered states induced by the frontier-based curriculum.

In contrast, the novelty-bonus baseline explores aggressively early on but rapidly collapses toward the shortest path between the start and the goal, leading to limited coverage of alternative routes and peripheral states.

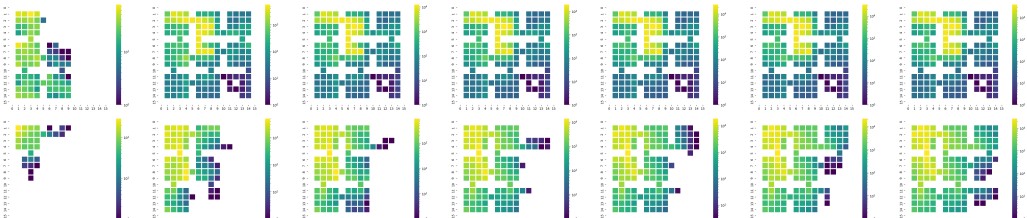

Figure 11: State visitations in NineRoomsLocked for Novelty bonuses (top) and SIERL (bottom) throughout training training, at 2k, 100k, 200k, 300k, 400k, 500k, and 600k steps. Agent starts in the top left corner and the goal is in the center of the central room.

### D.5 COMPARISON WITH MEGA EXPLORATION

MEGA (Maximum Entropy Gain Exploration) (Pitis et al.) is an exploration strategy that defines exploration problem as distribution matching of achieved and encourages the agent to systematically expand its knowledge by maximizing the entropy of its achieved goal distribution. Instead of relying on random exploration or external rewards, the agent deliberately sets goals from low density regions of achieved goal distribution. Instead of directly setting random goals that might not be reachable, by successfully returning to these states and then exploring locally, MEGA allows to continually increase entropy of achieved goal distribution, creating an automatic curriculum that bridges the gap between the starting state and distant, difficult task goals.

MEGA's desired behavior is fundamentally close to that of SIERL, albeit with a different sub-goal selection strategy. In MEGA, The agent looks at all the goals it has successfully reached in the past (stored in the replay buffer) and estimates how "crowded" or common each goal is using a density model (i.e., a Kernel Density Estimator). Thus, MEGA's frontier sampling involves selecting the "most novel" goals with the lowest density. In addition, OMEGA extends MEGA by blending its goal setting using uniformly random-goals, as MEGA's original goal distribution matching become more tractable.

Since MEGA relies on a KL divergence estimator which becomes increasingly demanding as the number of experiences increases, it is a more computationally intensive method. However, aside from the specific density estimation used, MEGA can be effectively implemented as a special case of SIERL's by adjusting hyperparameters accordingly. By setting the familiarity threshold to $F_\pi^{\text{thr}} = 1$, in order to sample sub-goals from the whole replay buffer, the novelty exponent $w_n = -1$ to prioritize novelty instead, and the path weights to $w_c = w_g = 0$ to make the selection agnostic to path cost estimates, we obtain an algorithm that closely emulates MEGA. To compare it with SIERL, we evaluated its performance on the DoorKey environment in Figure 10. It's performance matches closely that of SIERL with a slightly higher noise in main-goal success.

