# OpenReview forum: "Search Inspired Exploration for Reinforcement Learning"
_ICLR.cc/2026/Conference — Submitted to ICLR 2026_

### Official Review · Reviewer_rsWV · 2025-10-23

**Soundness:** 2
**Presentation:** 2
**Contribution:** 2
**Rating:** 4
**Confidence:** 5

**Summary:**

The paper proposes Search Inspired Exploration for RL (SIERL), a goal-conditioned method that expands a frontier of explored state-action pairs and selects sub-goals via a priority score that balances novelty with the learned estimates of $Q$-values. Training alternates between two steps of reaching a sampled sub-goal and pursuing the main task goal. Experiments on customized MiniGrid Hallway variants, BugTrap, and FourRooms environments show that SIERL performs competitively or better than HER, novelty bonuses, and other baselines.

**Strengths:**

* The framing of exploration as a search problem that progressively grows a frontier of reachable states is elegant and aligns with the intuition of structured exploration.
* The two-phased training approach naturally induces a curriculum from easy-to-reach subgoals toward the main goal. The ablations also demonstrate that removing the frontier or switching the components harms the performance of the agent.
* On discrete navigation tasks, SIERL reliably reaches main goals and achieves notably higher success on arbitrary goals, suggesting improved general exploration and goal generalization compared to discussed baselines.

**Weaknesses:**

The paper has the following main issues:
* Ambiguity and potential bias in $Q$-based cost terms
* Over-reliance on the environment geometry
* Limited metrics and hyperparameter selection

Issue 1:
The paper introduces cost-to-come and cost-to-go terms derived from the learned $Q$-values, but never clearly defines how these are estimated or trained. It is unclear whether $Q$ is defined over $(s, a, g)$ or $(s, g)$ and how maximization over actions is done. Also, how do these values interact with the alternating exploration phases? Since the same $Q$-values guide both sub-goal selection and policy training, the method may reinforce optimistic or inaccurate estimates.

Issue 2:
All the environments considered in the paper have their main goal at the farthest reachable location, making subgoal generation naturally align with distance. If the main goal is not at the farthest position (e.g., the main goal lies in the top-right corner for FourRooms environment), how will things change? Would the frontier still expand outward, or would it focus on goal-directed behavior? Demonstrating the robustness to different main goal placements or task geometries would test whether SIERL truly adapts its exploration frontier.

Issue 3:
In Figure 2, it’s not entirely clear whether the shaded regions represent confidence intervals or variance. Since the shaded areas overlap quite a bit, it’s difficult to draw strong conclusions about performance differences. Using only five random seeds may also limit the statistical confidence in the results, perhaps increasing the number of seeds or reporting statistical tests could help clarify the trends.
It also looks like several hyperparameters (such as phase horizons, familiarity thresholds, and percentile cutoffs) were manually tuned for each environment. It would be helpful if the paper could explain how these hyperparameters were selected. For example, whether they were tuned on validation environments or through a general heuristic, it will help to better understand the fairness and reproducibility of the comparisons.

Since the environments considered are discrete, including coverage metrics would offer a clearer, more quantitative view of each method’s exploration capability. Since the paper discusses other exploration strategies such as pseudo-count and frontier-based methods (RND, Go-Explore, LEAF, TLDR), comparing them using this metric would further strengthen the evaluation.

**Questions:**

* What was the reason for choosing deterministic environments? Could SIERL scale to stochastic settings, and would that affect frontier expansion?
* How would the frontier behave if the main goal is not at the farthest location?
* Could you provide a small experiment or visualization validating the curriculum aspect?

Addressing these would greatly improve both the clarity and empirical credibility of the paper.

---
*Minor comments*:
* Define the $\operatorname{softmin}$  in Section 4.3 and specify all score weights explicitly. Define $z(\cdot)$, $\sigma(\cdot)$ and vector weights once.
* The BugTrap environment looks similar to the hallway environment, so what’s the main reason for the significant difference in performance in this environment?
* Proofread for minor formatting issues (cost free,meaning Pg. 4)
* Since SIERL doesn’t reward raw novelty, it can plausibly avoid the noisy-TV problem. Highlighting this point will be nice to the paper. You can further test it in a stochastic or distractor-rich environment.

---

> ### Author Response · Authors · 2025-12-04
> **Official Comment by Authors (1/4)**
>
> We thank the reviewer for their thoughtful review. We are glad they found our framing of exploration as a search problem to be elegant and appreciated that the two-phase training naturally induces a curriculum from easy-to-reach sub-goals toward the main goal. We are also pleased that the reviewer noted our improved general exploration and goal generalization capabilities compared to baselines. To address your concerns regarding the definition of Q-based cost terms, the robustness of the method to different goal geometries, and the statistical confidence of our results, we provide the following clarifications and additional details.
>
> > Issue 1: The paper introduces cost-to-come and cost-to-go terms derived from the learned Q-values, but never clearly defines how these are estimated or trained. It is unclear whether Q is defined over (s, a, g) or (s, g) and how maximization over actions is done. Also, how do these values interact with the alternating exploration phases? Since the same Q-values guide both sub-goal selection and policy training, the method may reinforce optimistic or inaccurate estimates.
>
> SIERL is using Q-learning with ε-Greedy exploration, and the value function is defined over (s, g). The cost estimates are obtained directly from the learned values, requiring no separate estimator. We would like to emphasize that these estimates’ role is to partly influence the sub-goal selection for phase 1. Thus, more concretely:
>  - They are useful for relative comparison and prioritizing states from the available frontier. Thus, any bias that affects all the states equally does not affect the method’s performance. Much less so with optimistic biasing, for the same reason an optimistic heuristic is useful in e.g. the A* search.
>  - In the beginning, when the Q-values are the least accurate, the bias introduced by them has limited effect on the prioritization since novelty is still the main decisive factor for scoring the candidates. Additionally, the frontier itself is still small, and expansion in any direction is potentially useful.
>  - In addition to the above, over-reliance on poorly estimated Q-values is also limited thanks to the early switching, as if the agent is incapable of reaching a given subgoal through familiar states, it will be switched and in a new episode a new subgoal is chosen.
>
> > Issue 2: All the environments considered in the paper have their main goal at the farthest reachable location, making subgoal generation naturally align with distance. If the main goal is not at the farthest position (e.g., the main goal lies in the top-right corner for FourRooms environment), how will things change? Would the frontier still expand outward, or would it focus on goal-directed behavior? Demonstrating the robustness to different main goal placements or task geometries would test whether SIERL truly adapts its exploration frontier.
>
> We thank the reviewer for this question. When the main goal is not the farthest reachable state, the frontier continues to expand as long as novel states remain accessible. At the start of each episode, the agent may first pursue a frontier subgoal even if this temporarily moves it away from the main goal, after which it refocuses on the main goal. This does not hinder learning, as both the policy and all collected transitions are goal-conditioned. As a result, the set of mastered states expands outward (geometrically resembling an ellipse with the initial state and the main goal as foci), which improves random-goal generalization. At the same time, the diversity of states from which the agent learns to reach the main goal increases, strengthening generalization to that goal as well.
>
> This effect is visible in Figure 11, where the agent starts in the top-left corner and the goal is placed in the central room. After the agent first reaches the main goal (around 100k steps), the frontier continues to expand, and regions farther from both the initial state and the goal are visited less frequently. Additional coverage analyses are provided in Appendix D.4.

---

> ### Author Response · Authors · 2025-12-04
> **Official Comment by Authors (2/4)**
>
> > Issue 3: In Figure 2, it’s not entirely clear whether the shaded regions represent confidence intervals or variance. Since the shaded areas overlap quite a bit, it’s difficult to draw strong conclusions about performance differences. Using only five random seeds may also limit the statistical confidence in the results, perhaps increasing the number of seeds or reporting statistical tests could help clarify the trends. It also looks like several hyperparameters (such as phase horizons, familiarity thresholds, and percentile cutoffs) were manually tuned for each environment. It would be helpful if the paper could explain how these hyperparameters were selected. For example, whether they were tuned on validation environments or through a general heuristic, it will help to better understand the fairness and reproducibility of the comparisons.
>
> We thank the reviewer for this useful comment. We have now increased the number of seeds of all the main experiments and ablation studies from 5 to 10 in the new revision of the paper, to provide results with higher statistical confidence. Shaded regions are representing the standard error of the mean, as is  and this is mentioned in the original manuscript, near the end of par.1 in Subsection 5.1.
>
> We agree that there are several environment-depended hyperparameters, however we would like to point out that almost all of the hyperparameters remained constant across environments, with sole exception of the familiarity threshold, which was set to a limited range of values:
>
> | Environment | $F_\mathrm{thr}$ |
> | --- | --- |
> | Hallway 2-steps long | 0.9 |
> | Hallway 4-steps long, FourRooms | 0.8 |
> | Hallway 6-steps long | 0.95 |
> | BugTrap | 0.7 |
>
> The familiarity threshold controls the tradeoff between mastering already seen subgoals and practicing new subgoals. Values of familiarity closer to 1 lead to slower but steadier learning. Smaller values favour more aggressive exploration, but if the environment is constrained, it might harm the learning. The intuition behind these values in the table is that in larger and more open environments (e.g. BugTrap), a lower familiarity threshold allows for faster frontier propagation and new experience collection, since mastery there is not critical as there can exist a lot more possible trajectories to reach the frontier. On the other hand, in the narrower (more constrained) environments (e.g. long Hallway) the trajectory familiarity would generally need higher values (as the agent is forced to retrace a smaller number of available paths towards the frontier), requiring a higher threshold value in order to focus on the comparably less familiar states.
>
> Furthermore, we have added an Appendix B containing the hyperparameters used for all environments in our experiments. We investigated sensitivity to:
>
>  - Path weights
>  - Novelty weight (exponent)
>  - Familiarity threshold
>  - Softmin temperature
>
> From these results, it can be seen that there is a small impact on adjusting the weights, a moderate impact from softmin temperature, when set to a much larger value, and a comparably higher impact from the familiarity threshold. Diminishing performance when using a higher softmin temperature was expected, since it results in a "softer" distribution and more uniformly random prioritization, nullifying the method’s benefits. A too high familiarity threshold would considerably slow down the frontier expansion, and the solution-trajectory discovery, while maintaining a more stable random-goal success rate. A much lower threshold would lead to selecting sub-goals by more aggressively pursuing novelty, and spending less time “practicing” in the well-known region, which significantly reduced the random-goal success rate (the main-goal success was minimally impacted).

---

> ### Author Response · Authors · 2025-12-04
> **Official Comment by Authors (3/4)**
>
> > Since the environments considered are discrete, including coverage metrics would offer a clearer, more quantitative view of each method’s exploration capability. Since the paper discusses other exploration strategies such as pseudo-count and frontier-based methods (RND, Go-Explore, LEAF, TLDR), comparing them using this metric would further strengthen the evaluation.
>
> Thank you for this suggestion. We included coverage metrics in the Figure 11, visualising the visitation-counts for SIERL compared to other methods that are based on novelty-bonuses. It is visible that SIERL has significantly more uniform coverage of regions that cover approximately an ellipse with start and goal states as foci. For baseline, we have indeed used count-based novelty bonuses [1] with visitation-counts. There was no need for hashing or pseudo-counts since the state and action space is discrete, and plain counts can be used instead
>
> We would like to point out that, at a high level, all such methods (pseudocounts, intrinsic curiosity modules, RND, etc.) define an intrinsic reward bonus that is high if the current state is different from the previous states visited by the agent, and low if it is similar [2]. These methods approximate the novelty, each with a different approach and to a different degree. Using counts (when possible) provides the most precise way to quantify and reward novelty, compared to e.g. approximating surprise with RND. For this reason, and given that count-based novelty bonuses [1] is a well-performing representative from this family, for discrete state-action settings, we used this method as an upper bound for all of them.
>
> > What was the reason for choosing deterministic environments? Could SIERL scale to stochastic settings, and would that affect frontier expansion?
>
> We thank the reviewer for this useful suggestion. We have now included experiments with stochastic setting in Appendix D.2. Specifically, we used “slippery actions” where choosing to take the action e.g. “up” has 80% chance of being “up”, 10% chance of being “right”, and 10% chance of being “left”.  In this setting, SIERL outperforms all baselines. We observed that baseline methods experience a substantial degradation in both main-goal success and general exploration under stochasticity, whereas SIERL is affected only mildly: aside from a small reduction in main-goal success rate, overall performance and exploratory coverage remain stable. This suggests that SIERL’s frontier-based curriculum provides inherent robustness to transition noise.
>
> > How would the frontier behave if the main goal is not at the farthest location?
>
> (Answered above, in Issue 1.)
>
> > Could you provide a small experiment or visualization validating the curriculum aspect?
>
> We added an experiment in the Appendix D.4 that includes the coverage analysis of the explored state-space. Under SIERL, coverage expands outward in a structured manner. After the agent first reaches the main goal (around 100k steps), the frontier continues to grow, and regions farther from both the initial position and the goal receive progressively fewer visits. This reflects the incremental broadening of mastered states induced by the frontier-based curriculum.
>
> ---
>
> [1]: Tang, H., Houthooft, R., Foote, D., Stooke, A., Xi Chen, O., Duan, Y., Schulman, J., DeTurck, F., & Abbeel, P. (2017). #Exploration: A Study of Count-Based Exploration for Deep Reinforcement Learning. Advances in Neural Information Processing Systems, 30.
>
> [2]: Henaff, M., Raileanu, R., Jiang, M., & Rocktäschel, T. (2022). Exploration via Elliptical Episodic Bonuses. Advances in Neural Information Processing Systems, 35, 37631–37646.

---

> ### Author Response · Authors · 2025-12-04
> **Official Comment by Authors (1/4)**
>
> > The BugTrap environment looks similar to the hallway environment, so what’s the main reason for the significant difference in performance in this environment?
>
> The main difference between BugTrap and Hallway is that Hallway contains slippery tiles that bring the agent effectively to the beginning if it steps on them. This makes it required to perform the exact sequence of the going right actions to reach the goal. Contrary, in BugTrap the agent can make wrong moves and correct itself without significant harm and there are many more paths that solve the problem.
> Throughout learning SIERL learns to reach random goals in a gradually expanding part of the (now easily reachable) state space, even without/before learning to reach the “tougher” main goal. Thus, as designed to, it manages to reach intermediate states lying between the main goal and start, up until the point the main goal itself becomes part of those learned goals.
>
> This is much more apparent in more vast and wide open spaces (such as in FourRooms and BugTrap) as it takes more time to expand the frontier and cover the distance between start and main goal, compared to the narrower environments (Hallways). Eventually, SIERL manages to learn the main-goal much later while maintaining a high random-goal success rate, however maintaining a stable success rate on both for a much larger number of states in cases like this might be limited by the estimator’s (network’s) capacity itself.
>
> > Since SIERL doesn’t reward raw novelty, it can plausibly avoid the noisy-TV problem. Highlighting this point will be nice to the paper. You can further test it in a stochastic or distractor-rich environment.
>
> This is a great suggestion, we tested it on a stochastic environment and performance gain of using SIERL indeed is even better. Please find more details in the appendix D.2
>
> We thank the reviewer for their insightful suggestions, particularly regarding the Q-value estimation and the impact of environment geometry, as well as for helping improve our work.

---

### Official Review · Reviewer_Yhhq · 2025-10-31

**Soundness:** 3
**Presentation:** 2
**Contribution:** 1
**Rating:** 2
**Confidence:** 4

**Summary:**

The paper studies exploration in sparse-reward reinforcement learning (RL).
The paper proposes a method that lets the agent attempt intermediate subgoals which are (1) at the frontier of its capabilities and (2) in the right "direction" as estimated by the agent's learned Q-values.
The paper then evaluates this method extensively in low-dimensional discrete environments

**Strengths:**

The paper studies an important and challenging problem: effective exploration without dense rewards.
The paper studies the proposed method systematically in reasonable lower-dimensional discrete environments.

**Weaknesses:**

The main weakness of the paper is that the proposed method is extremely closely related to [1]. [1] proposed a method, which also selects intermediate subgoals from a reachable frontier and balancing exploration-exploitation by balancing cost-to-come and cost-to-go. Moreover, [1] evaluates this method in high-dimensional navigation & manipulation tasks which closely resemble the environments studied in this paper.

I think this work has the potential to contribute to the field, but in my view this paper would need to systematically compare against [1].
To sufficiently contrast with [1] and systematically evaluate extensions to [1], the paper would most likely have to be rewritten substantially.
It might also be informative to evaluate against other methods to automatically select subgoals as referenced in [1], such as the common MEGA [2].
In the current state, I unfortunately cannot recommend the paper for acceptance.

Additional points:
* The proposed method seems a bit more complex than [1] and it would be interesting to analyze which extensions are necessary. The methods section is also hard to follow in some places. It might be useful to include a more detailed algorithm box within the main paper.
* It would be good to systematically ablate the hyperparameters such as $w_n, w_c, w_g$ and $F_{\pi}^{thr}$.

[1]: Diaz-Bone et al., DISCOVER: Automated Curricula for Sparse-Reward Reinforcement Learning. https://arxiv.org/pdf/2505.19850
[2]: Pitis et al., Maximum entropy gain exploration for long horizon multi-goal reinforcement learning. https://arxiv.org/pdf/2007.02832

**Questions:**

See above.

---

> ### Author Response · Authors · 2025-12-04
> **Official Comment by Authors (1/2)**
>
> We thank the reviewer for their review. We are glad they recognized that our work tackles an important and challenging problem in effective exploration without dense rewards and that we conducted a systematic study of the method in reasonable environments. To address your primary concern regarding the relationship between our method and prior work such as DISCOVER and MEGA, as well as your feedback regarding the clarity of the methods section, we clarify the distinct contributions of SIERL and provide further comparisons below.
>
> > The main weakness of the paper is that the proposed method is extremely closely related to [1]. [1] proposed a method, which also selects intermediate subgoals from a reachable frontier and balances exploration-exploitation by balancing cost-to-come and cost-to-go. Moreover, [1] evaluates this method in high-dimensional navigation & manipulation tasks which closely resemble the environments studied in this paper.
>
> We thank the reviewer for pointing us to this valuable work that we were unaware of. During our work on this paper (ICLR submission deadline was 24 Sep 2025), we were not aware of DISCOVER, which is being currently presented at NeurIPS 2025 and was only publicly available on arxiv on 29. May 2025. DISCOVER is indeed using a similar Go-Explore-style two phase approach and concept of prioritization (although differently formulated) based on notions of novelty and promisingness (or “relevance” in their words) however, we noted a few significant differences with our work:
>
>  - SIERL first obtains a frontier by filtering the experiences, which is then prioritized and our biggest contribution is on elegant formulation of the frontier based on our defined familiarity score. DISCOVER does not use such a notion, but relies on the variance of an ensemble’s value estimations to measure uncertainty.
>  - Importantly, DISCOVER is using HER in its base implementation, which adds a non-negligible additional benefit to a goal-conditioned method, regardless of the method’s unique components.
>  - In DISCOVER’s terms, SIERL’s achievability is coupled with relevance, while both are being masked by novelty.
>  - Lastly, DISCOVER is evaluated using Test-Time Training (TTT). In TTT an agent is trained specifically for the target task at test-time, thus the learned policy itself is never evaluated for reaching the main goal from the initial state. SIERL was designed to produce a policy/value function capable of this, with the addition of generalizing and being able to reach other desired goals within the explored region.
>
> > I think this work has the potential to contribute to the field, but in my view this paper would need to systematically compare against [1]. To sufficiently contrast with [1] and systematically evaluate extensions to [1], the paper would most likely have to be rewritten substantially. It might also be informative to evaluate against other methods to automatically select subgoals as referenced in [1], such as the common MEGA [2]. In the current state, I unfortunately cannot recommend the paper for acceptance.
>
> We would argue that DISCOVER [1] should not constitute the baseline as it is just being presented at NeurIPS during the rebuttal period. We added comparison to MEGA [2] in the Appendix D.5, together with details on our implementation. In addition to the points presented in our previous answer about DISCOVER, MEGA exhibits similarities, albeit with the following key differences
>
>  - MEGA itself explores“blindly” by:
>     1. candidates from the buffer.
>     2. Filtering for achievability, using a Q-value cutoff (which is adjusted based on the agent's current goal achievement during exploration)
>     3. Prioritizing by Minimum Density, i.e. choosing the least visited state.
>  - OMEGA extends this by:
>     1. Calculating the "distance" or disconnect (KL divergence) between the Desired Goal distribution and the historical Achieved Goal distribution.
>     2. If that is too large: exploring using MEGA; else: exploiting and pursuing the main goal for the whole episode length.
>     3. Having no notion of promisingness for exploration, and can thus be aiming for irrelevant and hard/unreachable states.
>  - Lastly, as the authors noted, it is a computationally intensive method, relying on a KL divergence estimator which becomes increasingly demanding as the number of experiences increases.
>
> ---
>
> [1]: Diaz-Bone, L., Bagatella, M., Hübotter, J., & Krause, A. (2025). DISCOVER: Automated Curricula for Sparse-Reward Reinforcement Learning (No. arXiv:2505.19850). arXiv. https://doi.org/10.48550/arXiv.2505.19850
>
> [2]: Pitis, S., Chan, H., Zhao, S., Stadie, B., & Ba, J. (2020). Maximum entropy gain exploration for long horizon multi-goal reinforcement learning. Proceedings of the 37th International Conference on Machine Learning, 119, 7750–7761.

---

> ### Author Response · Authors · 2025-12-04
> **Official Comment by Authors (2/2)**
>
> > The proposed method seems a bit more complex than DISCOVER and it would be interesting to analyze which extensions are necessary. The methods section is also hard to follow in some places. It might be useful to include a more detailed algorithm box within the main paper.
>
> We would argue that the principle behind the method is simple and robust. We adapted the presentation in the Section 4.2 and 4.3 to improve comprehension. The ablation study (Section 5.3) showed the importance of all components in the algorithm. We additionally added the sensitivity analysis in Appendix D.1, that showed robustness and influence of hyperparameters.
>
> For a clear outline of the method, we would like to point the reader to the pseudocode in Appendix A, as well as the added Appendix B.1 with a more technical description of the frontier and experience management.
>
> > It would be good to systematically ablate the hyperparameters such as $w_\mathrm{n}$, $w_\mathrm{c}$, $w_\mathrm{g}$, and $F_\mathrm{thr}$.
>
> We thank the reviewer for this suggestion. We have run additional experiments, and additional hyperparameter sensitivity studies have been included in Appendix D.1. We investigated sensitivity to:
>
>  - Path weights
>  - Novelty weight (exponent)
>  - Familiarity threshold
>  - Softmin temperature
>
> From these results, it can be seen that there is a small impact on adjusting the weights, a moderate impact from softmin temperature, when set to a much larger value, and a comparably higher impact from the familiarity threshold. Diminishing performance when using a higher softmin temperature was expected, since it results in a "softer" distribution and more uniformly random prioritization, nullifying the benefits of our method. Too high a value for familiarity threshold would considerably slow down the frontier expansion, and the solution-trajectory discovery, while maintaining a more stable random-goal success rate. A much lower threshold would lead to selecting sub-goals by more aggressively pursuing novelty, and spending less time “practicing” in the well-known region, which significantly reduced the random-goal success rate (the main-goal success was minimally impacted), unless the environment is too hard that such aggressive expansion hurts learning process.
>
> We thank the reviewer for their valuable comments regarding the related work and for helping improve our work.

---

### Official Review · Reviewer_ZjhL · 2025-10-31

**Soundness:** 2
**Presentation:** 3
**Contribution:** 2
**Rating:** 6
**Confidence:** 4

**Summary:**

The paper introduces SIERL (Search-Inspired Exploration for RL), a goal-conditioned exploration method that systematically expands a frontier of known state–action pairs and sets sub-goals from this frontier using a priority that blends (i) a familiarity/novelty filter and (ii) search-style cost-to-come and cost-to-go estimates derived from learned Q-values. Each episode alternates between two phases: (1) reach a selected frontier sub-goal to push the boundary outward, then (2) pursue the main task goal from that more informative starting point. Experiments on MiniGrid variants (Hallway, FourRooms, BugTrap) show higher main-goal success and better generalization to random goals than baselines; ablations confirm the importance of early switching, frontier filtering, and prioritization. Contributions include the frontier extraction + sub-goal selection mechanism, a Hallway benchmark controlling action-sequence difficulty, and an empirical study disentangling component effects.

**Strengths:**

- Introduces a frontier extraction + prioritization mechanism using a familiarity filter and softmin over cost-to-come/go estimates, giving a principled way to pick feasible, informative sub-goals rather than random waypoints.

- Provides clear algorithmic structure and pseudo-code (Algorithms 1–3), detailing phase switching (fixed horizons + probabilistic early switch), frontier maintenance, and sub-goal sampling, supporting faithful reimplementation.

- Method motivation and the exploration–exploitation dilemma in goal-MDPs with sparse rewards are clearly articulated before the two-phase strategy, making the design choices easy to follow.

**Weaknesses:**

- Overlap with classic goal-conditioned curricula (frontier expansion, sub-goals via reachability/cost heuristics) is high; the paper doesn’t sharply distinguish SIERL from HER-style relabeling or novelty/prioritized-goal sampling beyond the specific softmin priority and early-switch heuristic.

- The current implementation relies on visitation counts for novelty/familiarity, limiting applicability to continuous state–action spaces and making SIERL sensitive to discretization choices. The authors acknowledge this limitation and suggest pseudo-counts but do not evaluate them.

- Excluding both “too familiar” and “too novel” (s,a) pairs creates a narrow band on the frontier; while intuitive, its necessity vs. a simpler top-K novelty or Q-margin selector is not isolated.

**Questions:**

- Please write the exact priority function you use to select frontier sub-goals (including normalization/z-scoring, softmin temperature, and any novelty weights). How sensitive is performance to these constants? A small table of priors → success rates would help

- How exactly do you detect and maintain the frontier set (data structures, update frequency, de-duplication)? What are the tie-break rules when multiple candidates share the same priority, and how do you handle an empty or exploding frontier?

- You alternate between (A) reaching the frontier sub-goal and (B) pursuing the main goal, with a probabilistic “early switch.” What is the trigger, and how do you pick its probability/horizon? Ablate fixed vs. adaptive switching (e.g., based on estimated success probability or advantage).

- Are these derived directly from learned Q values or from a separate estimator? How do you mitigate bias when Q is poorly estimated early on? Please compare: (i) raw Q, (ii) value-ensemble uncertainty, and (iii) model-based short-rollout costs.

- Since counts don’t scale to continuous spaces, which pseudo-count or density proxy (e.g., kNN in feature space, RND density) works best for SIERL? Show results on a continuous maze (e.g., AntMaze) and analyze robustness to representation choice.

---

> ### Author Response · Authors · 2025-12-04
> **Official Comment by Authors (1/3)**
>
> We thank the reviewer for their constructive feedback and detailed assessment. We are glad they found that our frontier extraction and prioritization mechanism offers a principled way to select feasible and informative sub-goals. We also appreciate that the reviewer recognized the clarity of our algorithmic structure and pseudo-code, noting that it supports faithful reimplementation, and that the motivation regarding the exploration-exploitation dilemma was clearly articulated. To address your concerns, we have provided detailed responses below.
>
> > Overlap with classic goal-conditioned curricula (frontier expansion, sub-goals via reachability/cost heuristics) is high; the paper doesn’t sharply distinguish SIERL from HER-style relabeling or novelty/prioritized-goal sampling beyond the specific softmin priority and early-switch heuristic.
>
> We agree that this method shares  concepts from classic goal-conditional curricula methods; however, we would like to point out that the familiarity-based frontier filtering and novelty and promisingness based prioritization are novel contributions in this work. In addition, we’d like to clarify that in SIERL, we do not rely on relabeling like in HER. HER was only used as a baseline to compare to.
>
> > The current implementation relies on visitation counts for novelty/familiarity, limiting applicability to continuous state–action spaces and making SIERL sensitive to discretization choices. The authors acknowledge this limitation and suggest pseudo-counts but do not evaluate them.
>
> This is indeed a current limitation, and the most promising direction for future work on this method. After having verified the method’s effectiveness with the experiments presented in this work, we believe it will be valuable to use a novelty/familiarity suitable for continuous state-action spaces, and examine whether SIERL is likewise capable in these settings.
>
> > Excluding both “too familiar” and “too novel” (s,a) pairs creates a narrow band on the frontier; while intuitive, its necessity vs. a simpler top-K novelty or Q-margin selector is not isolated.
>
> We find this a valuable point as well. Indeed, we have singled out pure novelty pursuit and focused on a band filter to obtain a frontier. Examining the tradeoffs (or any lack thereof) between simpler implementations and the current one is a valid consideration.
>
> > Please write the exact priority function you use to select frontier sub-goals (including normalization/z-scoring, softmin temperature, and any novelty weights). How sensitive is performance to these constants? A small table of priors → success rates would help.
>
> Thank you for this comment. The whole process is described in subsections 4.2 and 4.3, respectively. In the updated version, we have slightly adjusted the presentation and now the equation in 4.3 shows clearly that the softmin is applied to the obtained frontier, and that the sub-goal is sampled out of those.
>
> The complete sub-goal selection process consists of (1) filtering the Replay Buffer (RB) to obtain the frontier state-action pairs, (2) assigning probabilities to those based on the novelty and promisingness scoring, and drawing a state-action from the distribution obtained. More specifically:
>
>  1. For the filtering: each visited state-action is assigned a familiarity score (with our defined metric), and the pairs with a familiarity score below the specified $F_\mathrm{thr}$) are excluded (others are treated as already mastered subgoals), presented in subsection 4.2.
>  2. The remaining candidates constitute the “frontier” of exploration and are assigned a score and a probability (by applying softmin to the set of scores) from which the sub-goal is sampled (subsection 4.3).
>
> Additionally, extensive hyperparameter sensitivity studies have been included in Appendix D.1. We investigated sensitivity to:
>  - Path weights
>  - Novelty weight (exponent)
>  - Familiarity threshold
>  - Softmin temperature
>
> From these results, it can be seen that there is a small impact on adjusting the weights $w_\mathrm{n}$, $w_\mathrm{c}$, $w_\mathrm{g}$, a moderate impact from softmin temperature, when set to a much larger value, and a comparably higher impact from the familiarity threshold. Diminishing performance when using a higher softmin temperature was expected, since it results in a "softer" distribution and more uniformly random prioritization, nullifying the method’s benefits. A too high familiarity threshold would considerably slow down the frontier expansion, and the solution-trajectory discovery, while maintaining a more stable random-goal success rate. A much lower threshold would lead to selecting sub-goals by more aggressively pursuing novelty, and spending less time “practicing” in the well-known region, which significantly reduced the random-goal success rate (the main-goal success was minimally impacted).

---

> ### Author Response · Authors · 2025-12-04
> **Official Comment by Authors (2/3)**
>
> > How exactly do you detect and maintain the frontier set (data structures, update frequency, de-duplication)? What are the tie-break rules when multiple candidates share the same priority, and how do you handle an empty or exploding frontier?
>
> The frontier set is obtained by filtering out the replay buffer, and the familiarity scores used for this are calculated once stored on each step. To reduce memory usage and array insertions/deletions we maintain a dictionary of these values with the state-actions as the indexes. For more specific details please refer to the added Appendix B.1.
>
> Regarding tie-breaking, after prioritizing the candidates with a score and assigning probabilities based on that, the tie-breaking is inherently handled by the sampling itself. Even if two candidates share the same score, they are simply assigned the same probability, and one is selected at random.
>
> The cases of an empty or exploding frontier have been taken into account implicitly. In the case of an empty frontier, we simply directly switch to the main goal, thus turning that episode into a typical main-goal pursuit. But this happens usually once the main goal is already reached. We agree that this was not clear in the paper, and we have added a clarification in subsection 4.2, par. 2. The case of an exploding frontier is being effectively avoided thanks to the early-termination during phase 1, while using a reasonably short phase 2 length. By limiting the number of new state-action insertions in the RB (compared to the revisiting of existing entries) we can ensure the agent is able to consistently improve its confidence in the region within the frontier, while expanding it controllably.
>
> > You alternate between (A) reaching the frontier sub-goal and (B) pursuing the main goal, with a probabilistic “early switch.” What is the trigger, and how do you pick its probability/horizon? Ablate fixed vs. adaptive switching (e.g., based on estimated success probability or advantage).
>
> Continuing along from our previous answer, the trigger is stepping on a completely novel state. The probability of switching to phase 2 is a fixed hyperparameter that we have kept fairly large in our experiments, in order to foster more systematic frontier expansion and limit uncontrollable wandering around of the agent before achieving mastery (as explained above). Please refer to the now added Appendix B for the hyperparameters we used.
> Future extensions might be using an adaptive switching based on the estimated success probability(it has been used successfully in methods such as in [1]).
>
> > Are these derived directly from learned Q-values or from a separate estimator? How do you mitigate bias when Q is poorly estimated early on? Please compare: (i) raw Q, (ii) value-ensemble uncertainty, and (iii) model-based short-rollout costs.
>
> Yes, the cost estimates are obtained directly from the learned Q-values, requiring no separate estimator. We would like to emphasize that these estimates’ role is to guide the sub-goal selection for phase 1 such that it is more goal-directed and they do not influence frontier selection itself. Thus, more concretely:
>
>  - They are useful for relative comparison and prioritizing states from the available frontier. Thus, any bias that affects all the states equally does not affect the method’s performance. Much less so with optimistic biasing, for the same reason an optimistic heuristic is useful in e.g. the A* search.
>
>  - In the beginning, when the Q-values are the least accurate, the bias introduced by them has limited effect on the prioritization since novelty is still the main decisive factor for scoring the candidates. Additionally, the frontier itself is still small, and expansion in any direction is potentially useful.
>
>  - In addition to the above, over-reliance on poorly estimated Q-values is also limited thanks to the early switching, as if the agent is incapable of reaching a given subgoal through familiar states, it will be switched and in a new episode a new subgoal is chosen.
>
> ---
>
> [1]: Patil, D., Rajendran, J., Berseth, G., & Chandar, S. (2023, October 13). Intelligent Switching for Reset-Free RL. The Twelfth International Conference on Learning Representations.

---

> ### Author Response · Authors · 2025-12-04
> **Official Comment by Authors (3/3)**
>
> > Since counts don’t scale to continuous spaces, which pseudo-count or density proxy (e.g., kNN in feature space, RND density) works best for SIERL? Show results on a continuous maze (e.g., AntMaze) and analyze robustness to representation choice.
>
> SIERL is agnostic to the specific form of the count approximation: it can use any proxy (e.g., pseudo-counts or density-based estimates) because the method is orthogonal to how novelty is computed and does not rely on novelty bonuses directly. It is only necessary to map the values of novelty to our used range [0,1]. This extension is a promising direction, but not feasible within the scope of the current submission. Evaluating a broad set of continuous-space density proxies and benchmarking on environments such as AntMaze is therefore left for follow-up work.
>
> We thank the reviewer for their valuable suggestions, as well as for helping improve our work.

---

### Official Review · Reviewer_qyhj · 2025-10-31

**Soundness:** 2
**Presentation:** 3
**Contribution:** 2
**Rating:** 2
**Confidence:** 4

**Summary:**

The paper proposes SIERL (Search-Inspired Exploration for RL), a goal-conditioned exploration scheme that maintains a frontier of candidate state–action sub-goals filtered by a familiarity/visit-count criterion. It also prioritizes sub-goals using a softmin over (novelty cost)×(weighted sum of cost-to-come & cost-to-go), and runs a two-phase episode schedule, which includes reach sub-goal then pursue the main goal with early-switching when encountering novel states. Experiments are in discrete MiniGrid-style worlds with metrics cover main-goal success and random-goal success.

**Strengths:**

1. The paper has clear mechanism and ablations. The frontier construction, softmin prioritization over (cost-to-come,cost-to-go), and early-switching are well specified and ablated.
2. This paper provides a framework that offers surriculum-like behavior without reward shaping, which extends its applicability. The method steers exploration by sub-goal scheduling rather than altering rewards (contrast with novelty bonuses/RND), which avoids reward hacking/noisy-TV pitfalls.
3. SIERL shows strong random-goal success, surpassing baselines.

**Weaknesses:**

1. The method relies on visit counts/familiarity and a discrete frontier. Despite the authors have noted this limitation and suggest pseudo-count extensions, the claims made in the paper should be narrowed to discrete, low-dimensional settings given the current status.
2. The pipeline relies on many environment-dependent hyperparameters—e.g., familiarity threshold , percentile cutoff, novelty exponent, weights components, horizons, timeout , and switch probability. Finding right hyperparameter itself becomes arguablely as difficult as define a curriculum.
​3. Evaluations stay on toy-like MiniGrid rooms. To strengthen impact, include standard hard suites (e.g., MultiRoom-N×, DoorKey-16×16) and at least a few ProcGen tasks.
4. Related work cites count-based families (hashing, pseudo-counts) and ICM, but the experiments don’t clearly include canonical count-based baselines (e.g., Tang et al. hashing counts; Bellemare/Ostrovski pseudo-counts) or ICM with matched tuning/compute. Please add one of them for a fair comparison. They are highly correlated to the paper's idea.

**Questions:**

1. How sensitive is SIERL to hyperparameter selctions? Provide sensitivity analyses and practical defaults/ranges.
2. Which “novelty bonus” implementations were used?
3. Can you demonstrate the scalability of SIERL? What's the task with highest dimensions that SIERL is able to solve?
4. For main-goal success, SIERL trails novelty bonuses in places, but for random-goal success it clearly outperforms on FourRooms and BugTrap. Why it is the case? If SIERL is doing well in random goal sucess, it should also finish the main goal perfectly. Is the random goal sampled here too simple compared to the actual goal?

---

> ### Author Response · Authors · 2025-12-04
> **Official Comment by Authors (1/3)**
>
> We thank the reviewer for their insightful comments. We are glad they found that our paper provides a clear mechanism with well-specified ablations, and appreciated that our framework offers curriculum-like behavior without the risks of reward shaping or reward hacking. We are also encouraged that the reviewer recognized SIERL's strong performance in random-goal success compared to baselines. To address your concerns, we provide detailed clarifications and additional context below.
>
> > The method relies on visit counts/familiarity and a discrete frontier. Despite that the authors have noted this limitation and suggest pseudo-count extensions, the claims made in the paper should be narrowed to discrete, low-dimensional settings given the current status.
>
> We have adjusted our claims accordingly in the revision and clarified the scope of the method. We also describe the follow-up steps required to extend the approach beyond the discrete setting. In Section 7, we further explain that the method does not depend on the mean of the novelty estimate and can operate with any externally provided estimate.
>
> > The pipeline relies on many environment-dependent hyperparameters—e.g., familiarity threshold, percentile cutoff, novelty exponent, weights components, horizons, timeout, and switch probability. Finding the right hyperparameter itself becomes arguably as difficult as defining a curriculum.
>
> We agree that there are several hyperparameters that influence the behaviour of our algorithm, however we would like to point out that almost all of the hyperparameters remained constant across all training environments, with sole exception of the familiarity threshold, which was set to a limited range of values:
>
> | Environment | $F_\mathrm{thr}$ |
> | --- | --- |
> | Hallway 6-steps long | 0.95 |
> | Hallway 2-steps long and 4-steps long | 0.9 |
> | FourRooms | 0.8 |
> | BugTrap | 0.7 |
>
> The familiarity threshold controls the tradeoff between mastering already seen subgoals and practicing new subgoals. Values of familiarity closer to 1 lead to slower but steadier learning. Smaller values favour more aggressive exploration, but if the environment is constrained, it might harm the learning. The intuition behind these values in the table is that in larger and more open environments (e.g. BugTrap), a lower familiarity threshold allows for faster frontier propagation and new experience collection, since mastery there is not critical as there can exist a lot more possible trajectories to reach the frontier. On the other hand, in the narrower (more constrained) environments (e.g. long Hallway) the trajectory familiarity would generally need higher values (as the agent is forced to retrace a smaller number of available paths towards the frontier), requiring a higher threshold value in order to focus on the comparably less familiar states.
>
> Furthermore, we have added an Appendix B containing the hyperparameters used for all environments in our experiments and intuition on hyperparameter adjustment. Additionally the sensitivity analysis (added in the Appendix D.1) shows that SIERL algorithm is rather robust to the values of hyperparameters like cost-to-come, cost-to-go, novelty exponent  and softmin temperature, as well as above mentioned influence of the familiarity threshold.
>
> > Evaluations stay in toy-like MiniGrid rooms. To strengthen impact, include standard hard suites (e.g., MultiRoom-N×, DoorKey-16×16) and at least a few ProcGen tasks.
>
> We thank the reviewer for this suggestion to strengthen our evaluation. To address this, we have conducted additional experiments on challenging 9 room variants of MultiRoom and DoorKey environments, which feature larger spatial structures and longer horizons. These results are now included in Appendix D.3.
>
> Our analysis shows that in these settings as well, while SIERL matches the performance of aggressive exploration baselines (novelty bonuses) on main goals, it achieves substantially better generalization on random goals. We have also added a coverage analysis (Figure 11) illustrating that SIERL’s frontier-based curriculum enables a structured expansion and broader mastery of the environment, whereas baselines tend to rapidly collapse exploration toward the shortest path.

---

> ### Author Response · Authors · 2025-12-04
> **Official Comment by Authors (2/3)**
>
> > Related work cites count-based families (hashing, pseudo-counts) and ICM, but the experiments don’t clearly include canonical count-based baselines (e.g., Tang et al. hashing counts; Bellemare/Ostrovski pseudo-counts) or ICM with matched tuning/compute. Please add one of them for a fair comparison. They are highly correlated to the paper's idea.
>
> We would like to point out that, at a high level, all such methods (pseudocounts, intrinsic curiosity modules, random network distillation error, etc.) define an intrinsic reward bonus based on some estimate of novelty, i.e. that is high if the current state is different from the previous states visited by the agent, and low if it is similar [1]. These methods approximate the novelty each using different approaches. Using counts (when possible, as in our case) provides the most precise way to quantify and reward novelty, compared to e.g. approximating surprise with RND. For this reason, and given that count-based novelty bonuses [2] is a very well-performing representative from this family, we used this method (in results represented as novelty bonuses)  with exact counts, as a fair  upper bound baseline for all of them. More details on baselines are included in the new Appendix C.
>
> > How sensitive is SIERL to hyperparameter selections? Provide sensitivity analyses and practical defaults/ranges.
>
> We would like to thank the reviewer for this question. We have run additional experiments and added additional hyperparameter sensitivity studies in Appendix D.1. We investigated sensitivity to:
>
>  - Path weights
>  - Novelty weight (exponent)
>  - Familiarity threshold
>  - Softmin temperature
>
> SIERL showed robust performance. From these results, it can be seen that there is a rather small impact of adjusting the weights $w_\mathrm{n}$, $w_\mathrm{c}$, $w_\mathrm{g}$, a moderate impact from softmin temperature, when set to a much larger value, and a comparably higher impact from the familiarity threshold. Diminishing performance when using a higher softmin temperature was expected, since it results in a "softer" distribution and more uniformly random prioritization, nullifying the benefits of our method. Too high a value for familiarity threshold would considerably slow down the frontier expansion, and the solution-trajectory discovery, while maintaining a more stable random-goal success rate. A much lower threshold would lead to selecting sub-goals by more aggressively pursuing novelty, and spending less time “practicing” in the well-known region, which significantly reduced the random-goal success rate (the main-goal success was minimally impacted), unless the environment is too hard that such aggressive expansion hurts learning process.
>
> > Which “novelty bonus” implementations were used?
>
> We used count-based novelty bonuses [1] with visitation-counts. There was no need for hashing or pseudo-counts since the state and action space is discrete, and plain counts can be used instead.
>
> We have now included this information along with descriptions of the other baselines in a newly added Appendix C.
>
> > Can you demonstrate the scalability of SIERL? What's the task with the highest dimensions that SIERL is able to solve?
>
> In this work our primary goal was to develop a deeper understanding of SIERL’s mechanisms, strengths, and current limitations rather than to maximize performance on very high-dimensional benchmarks. While we have not yet conducted a comprehensive scalability study, we provide preliminary evidence that SIERL can extend beyond the simplest grid-based tasks. In the Appendix D we include additional experiments on MultiRoom and DoorKey. These results suggest that SIERL can handle larger and more structured state spaces, although a full investigation of its scalability remains an important direction for future work.
>
> ---
>
> [1]: Henaff, M., Raileanu, R., Jiang, M., & Rocktäschel, T. (2022). Exploration via Elliptical Episodic Bonuses. Advances in Neural Information Processing Systems, 35, 37631–37646.
>
> [2]: Tang, H., Houthooft, R., Foote, D., Stooke, A., Xi Chen, O., Duan, Y., Schulman, J., DeTurck, F., & Abbeel, P. (2017). #Exploration: A Study of Count-Based Exploration for Deep Reinforcement Learning. Advances in Neural Information Processing Systems, 30.

---

> ### Author Response · Authors · 2025-12-04
> **Official Comment by Authors (3/3)**
>
> > For main-goal success, SIERL trails novelty bonuses in places, but for random-goal success it clearly outperforms on FourRooms and BugTrap. Why is it the case? If SIERL is doing well in random goal success, it should also finish the main goal perfectly. Is the random goal sampled here too simple compared to the actual goal?
>
> The random goals are indeed simpler in expectation. In our setup they are drawn uniformly from the state space, while the main goal coincidentally corresponds to one of the farthest and most demanding states to reach. During training, SIERL gradually expands the region of states it can reliably reach, and this process does not require mastering the most difficult state early on. As a result, SIERL can achieve strong random-goal performance even before it has learned to reach the main goal. This effect is more pronounced in large and open environments such as FourRooms and BugTrap, where the distance between the start and main goal is substantial and the expansion of the reachable frontier takes longer. In narrower environments such as Hallways, this gap is smaller, so the difference between random-goal and main-goal performance is less observable. In Appendix D we also include additional experiments where the goal is not the farthest state, as well as results on a DoorKey variant with locked doors, which further illustrate how SIERL behaves when goal difficulty and environment topology vary.
>
> We thank the reviewer for their valuable comments and feedback, particularly regarding the hyperparameter sensitivity analysis and our baselines.

---

### Author Response · Authors · 2025-12-04
**Global Response**

We thank all reviewers for their thorough evaluations and constructive feedback. Their comments significantly helped improve the clarity, completeness, and empirical strength of the paper.

In the revised version, we incorporated a substantial set of additions and clarifications directly motivated by the reviewers’ concerns:
 1. **Additional experiments and statistical robustness.**

    We increased all main and ablation experiments to 10 seeds, clarified the meaning of shaded regions, and added coverage-based evaluations.

 2. **Hyperparameter sensitivity.**

     We conducted and reported extensive sensitivity studies covering path-weight components, novelty exponent, familiarity threshold, and softmin temperature. We also clarified which hyperparameters are shared across environments and provided a full hyperparameter table in the appendix.

 3. **Additional environments and scalability tests.**

    We added results on DoorKey, MultiRoom; included stochastic-transition environments; and provided analyses of alternative goal placements. These additions demonstrate robustness to environment geometry and transition stochasticity.


 4. **Frontier extraction, priority formulation, and implementation details.**

    We clarified the frontier definition, cost-to-come and cost-to-go derivations from Q-values, tie-breaking behavior, and replay-buffer filtering. A detailed description and pseudocode have been added to the appendix, with improved exposition in Sections 4.2–4.3.


 5. **Baselines and related work.**

    We added new experiments and comparison to frontier-based and automatic subgoal-selection baselines (MEGA, OMEGA), expanded the description of baselines including count-based novelty-bonus methods (pseudocounts, intrinsic curiosity modules, RND, etc.), and clarified the connections and key differences to DISCOVER. We did not include a comparison to DISCOVER, as it is being presented at NeurIPS after our submission and was not available during our original experimental evaluation.

 6. **Curriculum visualization and qualitative analyses.**

    We added visualizations of frontier evolution and explored-state coverage to highlight the emergent curriculum induced by the method.


 7. **Limitations and scope.**

    We clarified that the current implementation relies on exact counts, narrowed the claims accordingly, and outlined how pseudo-count or density-based approximations could be integrated. We also emphasized that SIERL is agnostic to the specific novelty estimator, provided values can be mapped to the required range.
Across all points raised by the reviewers, we provided detailed responses and corresponding revisions. We believe the paper is now substantially stronger and that all major concerns have been addressed. Below are the detailed, point-by-point responses to each reviewer.
We appreciate the reviewers’ efforts and regret that limited interaction prevented further discussion. Should further clarification be useful, we would be glad to provide it.

---

### Meta-Review · Area_Chair_7kLk · 2026-01-09

**Summary:**

This paper introduces a method to facilitate exploration in sparse-reward environments by selecting sub-goals on the "frontier" of the agent's known state space. These sub-goals are prioritized using search-inspired estimates of cost-to-come and cost-to-go, derived from learned Q-values, while balancing their novelty. This two-phase strategy induces a natural learning curriculum that allows SIERL to outperform baselines like Hindsight Experience Replay in both achieving the main task and generalizing to arbitrary states. While currently implemented for discrete settings using visitation counts, the method demonstrates superior environment coverage and robustness to stochasticity without the risks of reward hacking.

**Reviewer Concerns:**

The authors addressed several reviewer concerns during the rebuttal by adding significant empirical data and technical clarifications. They successfully resolved issues regarding statistical robustness by increasing the number of seeds to 10 for all experiments and expanded the scope of their evaluation to include harder navigation tasks like MultiRoom and DoorKey. Concerns about hyperparameter sensitivity were largely mitigated through extensive new studies in the appendix covering cost weights, novelty exponents, and familiarity thresholds. Furthermore, they clarified the frontier definition and cost derivations from Q-values, which helped distinguish the method from related work like DISCOVER.

However, some concerns remain outstanding or were only partially addressed. The most significant is the scalability to continuous spaces, as the current implementation still relies on exact visitation counts for novelty, leaving density-based approximations as a "promising direction for future work". Some reviewers felt the overlap with classic curricula was high and that the method's complexity, particularly its reliance on several environment-dependent constants, remains a drawback. Finally, while results in stochastic environments were provided, the method has yet to be tested in high-dimensional manipulation tasks or distractor-rich settings to fully prove it avoids the "noisy-TV problem".

**Reviewer Scores:**

- **Reviewer qyhj:** Initially rejected the paper due to its reliance on discrete visitation counts and "toy-like" MiniGrid environments, while criticizing the difficult manual tuning of environment-dependent hyperparameters. The authors addressed these concerns by adding experiments on DoorKey and MultiRoom, providing a sensitivity analysis for hyperparameters, and increasing the statistical significance to 10 seeds. Given that the authors directly provided the more complex environments and statistical robustness requested, this reviewer would likely have increased their score to a 4, consider the remaining lack of a continuous-space evaluation.

- **Reviewer ZjhL:** Provided a marginal acceptance but noted high overlap with classic curricula and identified the lack of scalability to continuous spaces as a major limitation. In response, the authors clarified that their familiarity-based frontier filtering is a novel contribution, detailed their "lazy" frontier management system using hashable dictionaries, and provided the exact priority function used for sub-goal sampling. This reviewer would likely have maintained a 6, as the authors clarified the technical implementation details for faithful reproduction.

- **Reviewer Yhhq:** Recommended rejection based on the lack of comparison to closely related methods like DISCOVER and MEGA, as well as clarity issues in the methods section. The authors responded by adding a performance comparison to MEGA, arguing that DISCOVER was contemporaneous (presented at NeurIPS 2025), and improving the presentation of the algorithm's core mechanics in Sections 4.2 and 4.3. Since the authors addressed the specific missing baseline concerns and improved the method's exposition, this reviewer would likely have increased their score to a 4.

- **Reviewer rsWV:** Initially leaned toward rejection, citing ambiguity in Q-based cost terms and a limited evaluation focused on deterministic settings where the goal is always at the farthest location. The authors addressed these concerns by conducting new experiments with stochastic "slippery" actions, providing a coverage analysis (Figure 11) to visualize the curriculum, and demonstrating robustness when the goal is placed in non-extreme locations. This reviewer would likely have kept the score of 4, due to the lack of experiments in distractor-rich problems and a larger number of stochastic problems.

---

### Decision · Program_Chairs · 2026-01-26

Reject